

**Background Heterogeneity and Other Uncertainties in**
**Estimating Urban Methane Flux: Results from the**
**Indianapolis Flux (INFLUX) Experiment**
Nikolay V. Balashov[1], Kenneth J. Davis[1], Natasha L. Miles[1], Thomas Lauvaux[1,2],
Scott J. Richardson[1], Zachary R. Barkley[1], Timothy A. Bonin[3, 4]
[1]The Pennsylvania State University, University Park, Pennsylvania, USA
[2]Laboratory of Climate Sciences and Environment, Gif-sur-Yvette, France
[3]Cooperative Institute for Research in Environmental Sciences, Boulder, Colorado, USA
[4]Chemical Sciences Division, National Oceanic and Atmospheric Administration, Boulder, Colorado,
USA
**Abstract**
As natural gas extraction and use continues to increase, the need to quantify emissions of methane
($CH_4$), a powerful greenhouse gas, has grown. Large discrepancies in Indianapolis $CH_4$ emissions
have been observed when comparing inventory, aircraft mass-balance, and tower inverse modeling
estimates. Four years of continuous $CH_4$ mole fraction observations from a network of nine tower-
based cavity ring-down spectrometers measuring atmospheric $CH_4$ mole fractions at 39 to 136 m
above ground as part of the Indianapolis Flux Experiment (INFLUX) are utilized to investigate
four possible reasons for the abovementioned inconsistencies: (1) differences in definition of the
city domain, (2) a highly temporally variable and spatially non-uniform $CH_4$ background, (3)
temporal variability in $CH_4$ emissions, and (4) the presence of unknown $CH_4$ sources. Reducing
the Indianapolis urban domain size to be consistent with the inventory domain size decreases the
$CH_4$ emission estimation of the inverse modeling methodology by about 35% and thereby lessens
the discrepancy by bringing total city flux within an error range of one of the inventories.
Nevertheless, the inverse modeling estimate still remains about 40% higher than the inventory
value. Hourly urban background $CH_4$ mole fractions are shown to be heterogeneous and
temporally variable. Statistically significant, long-term biases in background mole fractions of 2-



5 ppb are found from single point observations from most wind directions. Random errors in
single point background mole fractions observed for a few hours are 20-30 ppb, but decrease
substantially when data are averaged over multiple days. Boundary layer budget estimates suggest
that Indianapolis $CH_4$ emissions did not change significantly when comparing 2014 to 2016.
However, it appears that $CH_4$ emissions may follow a diurnal cycle with daytime emissions (12-
16 LST) approximately twice as large as nighttime emissions (20-5 LST). The strongest $CH_4$
source in Indianapolis is the South Side Landfill. Other point sources, perhaps leaks from the
natural gas distribution system, are localized and transient, and do not appear to be a consistently
large source of $CH_4$ emissions in Indianapolis. Long-term averaging, spatially-extensive upwind
mole fraction observations, mesoscale atmospheric modeling of the regional emissions
environment, and careful treatment of the times of day and areal representation of emission
estimates is recommended for precise and accurate quantification of urban $CH_4$ emissions.

**1 Introduction**
From the beginning of the Industrial Revolution to 2011, atmospheric methane ($CH_4$) mole
fractions increased by a factor of 2.5 due to anthropogenic processes such as fossil fuel production,
waste management, and agricultural activities (Ciais et al., 2013). The increase in $CH_4$ is a concern
as it is a potent greenhouse gas (GHG) with a global warming potential 28-34 times greater than
that of $CO_2$ over a period of 100 years (Myhre et al., 2013). The magnitudes of component $CH_4$
sources, and the causes of variability in the global $CH_4$ budget, however, are not well understood
although there is some evidence that biogenic emissions may play an important role in the recent
$CH_4$ increases (Nisbet et al., 2016; Saunois et al., 2016). Improved understanding of $CH_4$
emissions is needed (National Academies of Sciences and Medicine, 2018).



53  In particular, the estimates of continental U.S. anthropogenic $CH_4$ emissions disagree.

54 Inventories from Environment Protection Agency (EPA) and Emissions Database for Global

55 Atmospheric Research (EDGAR) in 2008 reported emission values of 19.6 and 22.1 TgC $y^{-1}$

56 (Miller et al., 2013). However, top-down methodologies using aircraft and inverse modeling

57 framework found emission values of $32.4 \pm 4.5$ TgC $y^{-1}$ for 2004 and $33.4 \pm 1.4$ TgC $y^{-1}$ for 2007-

58 2008 respectively (Kort et al., 2008; Miller et al., 2013). Underestimation of natural gas (NG)

59 production and agricultural sources are possible reasons for this disagreement (Miller et al., 2013;

60 Brandt et al., 2014; Jeong et al., 2014). Efforts to reconcile GHGs emissions estimates using

61 atmospheric methods and inventory assessment have succeeded (Schuh et al., 2013; Zavala-Araiza

62 et al., 2015; Turnbull et al., 2019) when careful attention is given to the details of each method,

63 and targeted atmospheric data are available. A recent synthesis of emissions from the U.S. NG

64 supply chain demonstrated similar success and concluded that current inventory estimates of

65 emissions from U.S. NG production are too low and that emission from NG distribution is one of

66 the greatest remaining sources of uncertainty in the natural gas supply chain (Alvarez et al., 2018).

67  Due to the uncertainties in $CH_4$ emissions from NG distribution it is natural that urban

68 emissions are of interest as well. For example, studies indicate that ~60-100% of Boston $CH_4$

69 emissions are attributable to the NG distribution system (McKain et al., 2015; Hendrick et al.,

70 2016). Recent studies of urban $CH_4$ emissions indicate that the California Air Resources Board

71 (CARB) inventory tends to underestimate the actual $CH_4$ urban fluxes possibly due to fugitive

72 emissions that result from the large NG infrastructures common to the urban environments (Wunch

73 et al., 2009; Jeong et al., 2016; Jeong et al., 2017). The accuracy and precision of atmospheric

74 estimates of urban $CH_4$ emissions are limited by available atmospheric observations (Townsend-

75 Small et al., 2012), potential source magnitude variability with time (Jackson et al., 2014; Lamb



et al., 2016), errors in atmospheric transport modeling (Hendrick et al., 2016; Deng et al., 2017;
Sarmiento et al., 2017;), and complexity in atmospheric background conditions (Cambaliza et al.,
2014; Heimburger et al., 2017). In this work, detailed analysis of urban $CH_4$ mole fractions is
performed in the city of Indianapolis to better understand the aforementioned uncertainties of
urban $CH_4$ emissions.

The Indianapolis Flux Experiment (INFLUX; Davis et al., 2017) is a testbed for improving

quantification of urban GHGs emissions and their variability in space and time. INFLUX
(http://influx.psu.edu) is located in Indianapolis partly because of its isolation from other urban
centers and the flat Midwestern terrain. It includes a very dense GHGs monitoring network,
comprised of in situ aircraft measurements (Heimburger et al., 2017; Cambaliza et al., 2014), in
situ observations from communications towers using cavity ring-down spectroscopy (Richardson
et al., 2017; Miles et al., 2017), and automated flask sampling systems for quantification of a wide
variety of trace gases (Turnbull et al., 2015). Meteorological sensors include a Doppler lidar
providing continuous boundary layer depth and wind profiles, and tower-based eddy covariance
measurements of the fluxes of momentum, sensible and latent heat (Sarmiento et al., 2017). The
network is well suited for emissions estimates using top-down methods such as tower-based
inverse modeling (Lauvaux et al., 2016) and aircraft mass balance estimates (Cambaliza et al.,

2015).

Recently Lamb et al. (2016) compared Indianapolis $CH_4$ emissions estimates from

inventory, aircraft mass balances, and inverse modeling. The results revealed large mean
differences among the city fluxes estimated from these methods (Fig. 1). In general, the inventory
methods arrived at lower estimates of emissions compared to the atmospheric, or top-down
approaches. $CH_4$ fluxes calculated using the aircraft mass balance technique varied considerably



between flights, more than would be expected from propagation of errors of the component
measurements (Cambaliza et al., 2014; Lamb et al., 2016). The atmospheric inverse estimate was
significantly higher than the inventory and some of the aircraft-derived values.
Biogenic emissions from the city are dominated by a landfill close to downtown, and these
emissions are thought to be fairly well known. Uncertainty in emissions is driven by the
uncertainty in thermogenic emissions, which are hypothesized to emerge largely from the NG
distribution system (Mays et al., 2009; Cambaliza et al., 2015; Lamb et al., 2016). This uncertainty
has not yet been resolved. In this study, we explore potential explanations for the discrepancies in
$CH_4$ emissions estimates from Indianapolis and posit methods and recommendations for the study
of $CH_4$ emissions from other urban centers.
We examine four different potential explanations for the $CH_4$ flux discrepancies reported
in Lamb et al. (2016): (1) inconsistent geographic boundaries, (2) heterogeneity in the urban-scale
$CH_4$ background, (3) temporal variability in urban emissions, and (4) $CH_4$ sources that are not
accounted for in the inventories. Well-calibrated $CH_4$ sensors on the INFLUX tower network
(Miles et al., 2017) collected continuous $CH_4$ observations from 2013 to 2016 and provide a unique
opportunity to explore these issues.

**2 Methods**

**2.1 Experimental site**
This study uses data from a tower-based GHG observational network located in the city and
surrounding suburbs of Indianapolis, Indiana, in the Midwestern U.S. Prior studies have used
varying definitions for the region of Indianapolis (Cambaliza et al., 2015, Lamb et al., 2016). In





this work, we follow Gurney et al. (2012) and define Indianapolis as the area of Marion County.
The flat terrain of the region simplifies interpretation of the atmospheric transport. The land-
surface heterogeneity inherent in the urban environment (building roughness, spatial variations in
the surface energy balance) do have a modest influence on the flow within the city and the
boundary layer depth difference between the urban and rural areas (Sarmiento et al., 2017).

**2.2 INFLUX tower network**
The continuous GHG measurements from INFLUX are described in detail in Richardson et al.
(2017). The measurements were made using wavelength-scanned cavity ring down spectrometers
(CRDS, Picarro, Inc., models G2301, G2302, G2401, and G1301), installed at the base of existing
communications towers, with sampling tubes secured as high as possible on each tower (39 – 136
m above ground level (AGL), Miles et al., (2017)). A few towers also included measurements at
10 m AGL and one or two intermediate levels. While INFLUX tower in-situ measurements began
in September 2010, here we focus on the $CH_4$ measurements from 2013 – 2016. From June
through December 2012, there were two or three towers with operational $CH_4$ measurements. By
July 2013, five towers included measurements of $CH_4$, and throughout the majority of the years
2015 – 2016 there were eight INFLUX towers with $CH_4$ measurements (Fig. 2).

By May 2013, the inflow to all CRDS instruments was dried. Prior to deployment and

following any manufacturer repairs, the instruments were calibrated for slope and offset in the
Pennsylvania State University calibration laboratory (Richardson et al., 2017) using three to five
NOAA-calibrated tanks. At each site, one or two NOAA-calibrated tanks were sampled daily for
10 min as field offset calibration points. In this study we used hourly means of $CH_4$, which were



reported on the WMO X2004A scale. Flask to in-situ comparisons and round-robin style testing
indicated compatibility across the tower network of 0.6 ppb $CH_4$ (Richardson et al., 2017).

**2.3 Weather data**
Wind data was measured at the Indianapolis International Airport (KIND), Eagle Creek Airpark
(KEYE), and Shelbyville Municipal Airport (KGEZ). The data used are hourly values from the
Integrated Surface Dataset (ISD) (https://www.ncdc.noaa.gov/isd) and 5-minute values directly
from the Automated Surface Observing System (ASOS). A complete description of ASOS stations
is available at http://www.nws.noaa.gov/asos/pdfs/aum-toc.pdf. The accuracy of the wind speed
is ±1 m/s or 5% (whichever is greater) and the accuracy of the wind direction is 5 degrees when
the wind speed is ≥ 2.6 m/s. The anemometer is located about 10 meters AGL. The wind data
reported in ISD are given for a single point in time recorded within the last 10 minutes of an hour
and are closest to the value at the top of the hour.
The planetary boundary layer height (BLH) was determined from a Doppler lidar deployed
in Lawrence, IN, about 15 km to the northeast of downtown. The lidar is a Halo Streamline unit,
which was upgraded to have extended range capabilities in January 2016. The lidar continuously
performs a sequence of conical, vertical-slice, and staring scans to measure profiles of the mean
wind, turbulence, and relative aerosol backscatter. All of these measurements are combined using
a fuzzy-logic technique to automatically determine the BLH continuously every 20-min (Bonin et
al., 2018). The BLH is primarily determined from the turbulence measurements, but the wind and
aerosol profiles are also used to refine the BLH estimate. The BLHs are assigned quality-control
flags that can be used to identify times when the BLH is unreliable, such as when the air is
exceptionally clean, the BLH is below a minimum detectable height, or clouds and fog that



attenuate the lidar signal exist.  Additional details about the algorithm and the lidar operation for
the INFLUX project are provided in Bonin et al. (2018).  Doppler lidar measurements are available
at https://www.esrl.noaa.gov/csd/projects/influx/.

**2.4 CH$_4$ Sources**
Only a few known CH$_4$ point sources exist within Indianapolis (Cambaliza et al., 2015, Lamb et
al., 2016).  The Southside Landfill (SSLF), located near the center of the city, is the largest point
source in the city with emissions of about 28-45 mol/s, accounting for 22% to 63% of total Marion
County CH$_4$ emissions (Cambaliza et al., 2015; Maasakkers et al., 2016; Lamb et al., 2016).  Other
city point sources are comparatively small; the wastewater treatment facility located near SSLF
contributes approximately 4-10% to city CH$_4$ totals or about 3-7 mol/s, and the transmission-
distribution transfer station at Panhandle Eastern Pipeline (also known as a city gate and further in
this study abbreviated as PEP) is estimated to be about 0.5-1% or 1 mol/s.  The remaining CH$_4$
sources, mainly from NG and livestock, are considered to be diffuse sources and are not well
known.  Potential sources of emissions related to NG activities include gas regulation meters,
emissions from transmission and storage, and Compressed Natural Gas (CNG) fleets.  These
diffuse NG sources account for 21-69% or 20-64 mol/s of the city emissions (Cambaliza et al.,
2015; Maasakkers et. al. 2016; Lamb et al., 2016).  Livestock emissions for Marion County are
estimated to be around 3% or 1.5 mol/s.

**2.4 Urban methane background**
Both aircraft mass balance and inverse modeling methodologies rely on the accurate estimation of
the urban CH$_4$ enhancement relative to the urban CH$_4$ background in order to produce a reliable



flux estimate (Cambaliza et al., 2014; Lamb et al., 2016). The $CH_4$ mole fraction enhancement is
defined as,

$$C_{enhancement} = C_{downwind} - C_{bg} \qquad (1)$$

where $C_{downwind}$ is the $CH_4$ mole fraction measured downwind of the source and $C_{bg}$ is the $CH_4$
background mole fraction, which can be measured upwind of the source, but this is not necessary.
Background, as defined in this body of literature, is a mole fraction measurement that does not
contain the influence of the source of interest, but which is measured simultaneously. Because
choosing the background involves a degree of subjectivity (Cambaliza et al., 2014; Heimburger et
al., 2017) we consider how this choice may influence emission estimates and introduce error, both
random and systematic, using data from the INFLUX tower network.
Using tower network data from November, 2014 through the end of 2016, two $CH_4$
backgrounds are generated based on two different criteria. Both criteria identify a tower suitable
to serve as a background for each of the eight wind directions (N, NE, E, SE, S, SW, W, NW),
where an arc of 45° represents a direction (e.g. winds from N are between 337.5° and 22.5°).
Criterion 1 is based on the concept that the lowest $CH_4$ mole fraction measured at any given
time is not affected by the city sources and therefore is a viable approximation of the background
methane mole fractions outside of the city (Miles et al., 2017; Lauvaux et al., 2016). Given this
assumption, the tower with the lowest median of the $CH_4$ enhancement distribution (calculated by
assuming the lowest measurement among all towers at a given hour as a background) for each of
the wind directions over the November, 2014 through December, 2016 time period is chosen as a
background site (Miles et al., 2017). Criterion 2 requires that the tower is outside of Marion
County (outside of the city boundaries) and is not downwind of any known regional $CH_4$ source
(Fig. 3). For some wind directions, there are multiple towers that could qualify as a background;



we pick towers in such a manner that they are different for each criterion given a wind direction
in order to calculate the error associated with the use of different but acceptable backgrounds. The
towers used for both criteria and for each of the eight wind directions are displayed in Table 1.
Quantifying differences between these two backgrounds allows for an opportunity to better
understand the degree of uncertainty that exists in the Indianapolis background atmosphere.

**2.5 Frequency and bivariate polar plots**
Frequency and bivariate polar plots are used in this work to gain more knowledge regarding $CH_4$
background variability based on criteria 1 and 2, and to identify sources located within the city.
To generate these polar plots, we use the *openair* package (from R programming language) created
specifically for air quality data analysis (Carslaw and Ropkins, 2012). Bivariate and frequency
polar plots indicate the variability of a pollutant concentration at a receptor (such as an
observational tower) as a function of wind speed and wind direction, preferably measured at the
location of the receptor or within several kilometers of the receptor. The frequency polar plot is
generated by partitioning the $CH_4$ hourly data into the wind speed and direction bins of 1 m s$^{-1}$ and
10° respectively. To generate bivariate polar plots, wind components $u$ and $v$ are calculated for
hourly $CH_4$ concentration values, which are fitted to a surface using a Generalized Additive Model
(GAM) framework in the following way,

$$\sqrt{C} = \beta + s(u, v) + \epsilon \qquad (2)$$

where $C$ is the $CH_4$ mole fraction transformed by a square root to improve model diagnostics such
as a distribution of residuals, $\beta$ is mean of the response, $s$ is the isotropic smoothing function of
the wind components $u$ and $v$, and $\epsilon$ is the residual. For more details on the model see Carslaw
and Beevers (2013).




**2.6 Temporal Variability**
Temporal variability may play an important role in the quantification of urban methane emissions.
Lamb et al., (2016) suggested that temporal variability may partially explain the differences among
$CH_4$ flux estimates shown in Fig. 1.  If temporal variability of $CH_4$ emissions exists within the city,
disagreements in the $CH_4$ flux between studies could be attributed to differences in their sampling
period.  Because the INFLUX tower data at Indianapolis contain measurements at all hours of the
day over multiple years, we can utilize this dataset to better understand the temporal variability in
methane emissions in the city.
We apply a simplified atmospheric boundary layer budget, not to estimate precisely the
actual city emissions, but rather to evaluate temporal variability of the emissions.  We begin by
assuming $CH_4$ emissions $Q_a$ (mass per unit time per unit area) are not chemically active and are
constant over a distance $\Delta x$ spanning a significant portion of the city.  The next assumption is that
a $CH_4$ plume measured upwind of the city is well mixed within a layer of depth $z_i$.  We treat wind
speed $u$ as constant within the layer for every hour considered.  Given the above-mentioned
assumptions we can write a continuity equation describing mass conservation of $CH_4$
concentration $C$ within a box in the following fashion,

$$\Delta x z_i \frac{\partial C}{\partial t} = \Delta x Q_a + u z_i (C_b - C) + \Delta x \frac{\partial z_i}{\partial t}(C_a - C) \tag{3}$$

where $C_b$ is the $CH_4$ mole fraction upwind of the city (or background), and $C_a$ is the $CH_4$
concentration above the mixed layer (Hanna et al., 1982; Arya, 1999; Hiller et al., 2014).  The left
hand of the equation represents the change in $CH_4$ concentration with time, $\Delta x Q_a$ denotes a
constant $CH_4$ source over the distance $\Delta x$, $u z_i (C_b - C)$ indicates a change of $CH_4$ concentration
due to horizontal advection, and finally $\Delta x \frac{\partial z_i}{\partial t}(C_a - C)$ term accounts for the vertical advection



and encroachment processes that result from changing mixed layer height. By assuming steady
state conditions ($\frac{\partial C}{\partial t} = 0$ and $\frac{\partial z_i}{\partial t} = 0$), the equation can be simplified to

$$Q_a = \frac{u z_i (C - C_b)}{\Delta x} \qquad (4)$$

We use equation 4 to estimate hourly CH₄ emissions ($Q_a$) from Indianapolis (see
assumptions in the paragraph below), given hourly data of $z_i$ from the lidar positioned in the city,
wind speed from the local weather stations, and upwind ($C_b$) and downwind ($C$) CH₄
concentrations measured at towers 1, 8, and 13 (depending on wind direction) using data from
heights of 40 m, 41 m, and 87 m respectively (see Fig. 3).
The CH₄ concentrations are derived from CH₄ mole fractions by approximating average
molar density of dry air (in mol m⁻³) within the boundary layer for every hour of the day, where
variability of pressure with altitude is calculated using barometric formula and it is assumed that
temperature decreases with altitude by 6.5 K per kilometer. The hourly surface data for pressure
and temperature is taken from KIND weather station. The difference between concentrations
($C - C_b$) is instantaneous and not lagged, where $C_b$ represents air parcel entering the city and $C$
represents the same air parcel exiting the city (for more details see Turnbull et al., 2015). The CH₄
enhancements ($C - C_b$) are estimated for daytime by averaging observations spanning 12-16 LST
and for nighttime by averaging observations spanning 20-5 LST. These time periods are based on
lidar estimations of when on average $z_i$ varies the least. The day and night were required to contain
at least 3 and 9 hourly CH₄ values respectively for averaging to occur, otherwise the day/night is
eliminated. Observations when $z_i$ is below 100 m are not used to avoid the cases when
measurements from towers may be above the boundary layer. In order to better achieve the
assumption that the boundary layer is fully mixed (especially at night), all hours with wind speeds
below 4 m/s are eliminated (Van De Wiel., 2012). To approximate the emissions of the whole city



we need to know the approximate area of the city and the distance over which the plume is affected
by the city $CH_4$ sources. The area of the city is about 1024 $km^2$ (the area of Marion County) and
the length that plume traverses when it is over the city ranges from 32 to 35 km depending on
which downwind tower is used. We assume that $CH_4$ measurements at towers 8 and 13 are
representative of a vertically well-mixed city plume as the towers are located outside of the city
boundaries and allow for sufficient vertical mixing to occur. For S and SW wind directions tower
8 observations are used to represent downwind conditions with background observations coming
from towers 1 and 13, respectively (based on Criterion 1 shown in Table 1). For W wind direction,
tower 13 observations represent the downwind with background obtained from tower 1. The wind
direction is required to be sustained for at least 2 hours, otherwise the data point is eliminated.

**3 Results and discussion**
**3.1 City Boundaries**
A significant portion of $CH_4$ emissions across the U.S. can be characterized by numerous large
point sources scattered throughout the country rather than by broad areas of smaller enhancements
(Maasakkers et al., 2016). Because of this, the total emissions for a given domain can be very
sensitive to how that domain is defined. A small increase or a decrease in the domain area could
add or remove a large point source and significantly impact the total emissions defined within the
domain. This issue can be observed in prior studies of $CH_4$ emissions in Indianapolis, described
below.

In Fig. 3, two possible domains are identified (Lamb et al., 2016; Lauvaux et al., 2016) that

could be used for the evaluation of Indianapolis $CH_4$ emissions. The first domain is the whole area
shown in the figure enclosing both Indianapolis and places that lie outside of its boundaries. The
second domain is Marion County outlined with a green dashed line. It is assumed here that this





domain is much more representative of the actual Indianapolis municipal boundaries as this area
encompasses the majority of the urban development associated with the city of Indianapolis
(Gurney et al., 2012). The larger domain has three additional landfills that, based on the EPA
gridded inventory (Maasakkers et al., 2016), increase $CH_4$ emissions by about 50% when
compared to the smaller domain.

This issue became apparent when the emissions were calculated using an atmospheric

inversion model (Lamb et al., 2016; Lauvaux et al., 2016). The atmospheric inversion solved for
fluxes in domain 1, which significantly increased the estimated emissions in comparison with the
inventory values that were gathered mainly within Marion County (domain 2). When reduced to
domain 2, inverse modeling emission estimates decrease to 107 mol/s, which falls within an error
bar of Lamb et al. (2016) inventory estimate. This difference is significant and could at least
partially explain the discrepancy shown in Fig. 1 between the emission values from the inventories
and emission results from the inverse modeling. However, even the decreased inverse modeling
estimate is about 40% higher than the inventory.

The subject of the domain is also relevant for airborne mass balance flights because a priori

the magnitude and variability of background plume is unknown and could be easily influenced by
upwind sources. The issue of background is discussed further in the next section.

**3.2 Variability in Background Tower Mole Fraction**

Comparisons between Criterion 1 and Criterion 2 $CH_4$ mole fraction enhancements as a

function of wind direction are visualized using frequency and bivariate polar plots (Fig. 4). To
make the comparison as uniform as possible, only data from 12-16 LST are utilized (all hours are
inclusive), when the boundary layer is typically well-mixed (Bakwin et al., 1998). A lag 1





autocorrelation is found between 12-16 LST hours, i.e., the hourly afternoon data are correlated to
the next hour, but the correlation is not significant for samples separated by two hours or more.
Therefore, hours 13 and 15 LST are eliminated to satisfy the independence assumption for hourly
samples. Furthermore, we make an assumption that the data satisfy steady state conditions. If the
difference between consecutive hourly wind directions exceeds 30 degrees or the difference
between hours 16 and 12 LST exceeds 40 degrees, the day is eliminated. Days with average wind
speeds below 2 m/s are also eliminated due to slow transport (the transit time from tower 1 to
tower 8 is about 7 hours at a wind speed of 2 m/s).

Both backgrounds generally agree on the higher $CH_4$ originating from the SW, SE, and E

wind directions (Figs. 4c-f); however, the values themselves differ especially when winds are from
NW, SW, and SE. As the background difference plots indicate, there is noticeable variability in
the magnitudes of the $CH_4$ mole fraction background, where criterion 2, by design, typically has
higher background mole fractions. The background differences, at a given hour, suggest that the
$CH_4$ field enveloping the city is heterogeneous with differences between towers ranging from 0 to
over 20 ppb (Fig. 4g). Because large gradients in $CH_4$ background over the city could pose
challenges for flux estimations using top down methods such as inverse modeling and aircraft mass
balance, it is imperative to establish whether the background differences vary randomly or
systematically and how to choose a background to minimize these errors.

To further understand the nature of background variability we calculate the mean and

standard error of background hourly differences over November 2014 to December 2016 for each
of the eight wind directions mentioned in Table 1. The results are shown in Fig. 5. Systematic
bias is evident for the SE, S, SW, W, and NW wind sectors, whereas random error dominates N,





NE, and E wind directions.  Wind directions showing bias have mean biases ranging from 2 to 5
ppb, with values as large as 8 ppb falling within the range of two times the standard error.

Random errors in the mole fractions of background differences are also important and are

a function of the length of the data record.  We quantify the random error in the $CH_4$ background
mole fraction differences using the bootstrap method by randomly sampling 2 to 150 hours of the
background $CH_4$ differences for each of the wind directions with replacement (we make the
assumption that our differences are independent since we eliminated lag 1 autocorrelation from
the data). This sub-sampling experiment is repeated 5000 times (Efron and Tibshirani, 1986).  The
standard deviations of the mean (standard error) of the 5000 simulated differences are calculated
for each wind direction.  The resulting standard errors of the city $CH_4$ background, multiplied by
2 to represent the 95% confidence intervals, are shown as a function of the length of the data record
in Fig. 6.  All wind directions demonstrate that, as expected, the random error falls as the sample
size grows.  In general, 25 hourly samples of data reduce random errors by about 70%.  Using less
than 25 hourly samples to estimate emissions may result in random errors of $CH_4$ mole fractions
of 5 to 35 ppb.  Now we consider these random and systematic errors in the $CH_4$ background in
context of Indianapolis urban $CH_4$ emissions.

For Indianapolis, using INFLUX tower network, we estimated that depending on sample

size (number of hours sampled) and wind direction systematic and random errors of $CH_4$
background gradient across the city over 12-16 LST could vary from 0 to 5 ppb and from 5 to 35
ppb respectively.  Given that the average afternoon $CH_4$ enhancement of the city is around 8-12
ppb (section 3.3; Fig. 7; Cambaliza et al., 2015; Miles et al., 2017), the error on the estimated
emissions could be over 100% if the analysis does not approach the issue of background with
enough sampling.  For $CH_4$ sources with a significantly larger signal than their regional



background, the mentioned background variability becomes less impactful on results, but because
Indianapolis is a relatively small emitter of $CH_4$, the uncertainties due to background are
comparatively large. Our random error assessment suggests that the highly variable $CH_4$ emission
values of Indianapolis from the aircraft mass balance calculations (Fig. 1) are at least partially due
to the variability in the urban $CH_4$ background of Indianapolis.

**3.3. Temporal Variability**
Fig. 7 presents average $CH_4$ mole fraction enhancements and flux calculations (equation
4) at towers 8 and 13 for years 2014, 2016, and 2013-2016 (for the detailed methodology see
sections 2.6). The years of 2014 and 2016 are chosen for temporal comparison because they do
not contain major BLH data gaps. The error bars in the figure show the standard error multiplied
by 2 indicating 95% confidence interval of each average.
One of the more interesting features in the Fig. 7 is a day/night variability of $CH_4$ emissions
at Indianapolis. The most prominent example of this feature is found in Fig. 7c, where the
estimates for both years suggest that daytime emissions are at least twice as high as the emissions
at night. The decrease of the $CH_4$ emissions at night also appears in tower 13, but the errors are
too high in those estimates to make any definitive conclusions. A similar urban $CH_4$ emissions
diurnal variability is reported by Helfter et al. (2016) in their study of GHGs for London, UK,
where they attribute diurnal variation of the $CH_4$ emissions to the NG distribution network
activities, fugitive emissions from NG appliances, and to temperature-sensitive $CH_4$ emission
sources of biogenic origin (such as a landfill). Taylor et al. (2018) suggest that $CH_4$ emissions
from landfills exhibit a diurnal cycle with higher emissions in early afternoon and 30-40% lower
emissions at night.





With regard to yearly temporal variability we are only able to compare years 2014 and
2016 due to limited BLH data for other years. Results from both towers suggest that Indianapolis
$CH_4$ emissions did not change significantly.

**398    3.4 Sources**

Bottom-up emission inventories have difficulty tracking changes in sources over time. Our
continuous tower network observations can monitor temporal and spatial variability in sources of
$CH_4$ in Indianapolis. To do so we employ the aforementioned bivariate polar plots to verify known
sources and potentially identify unknown sources across the city. We compare two time periods,
2014-2015 (two full years) and 2016. Fig. 8 displays bivariate polar plots of $CH_4$ enhancements
using criterion 1 background at 9 INFLUX towers in Indianapolis over the two years of 2014 and
2015. Fig. 9 shows the same plot, but for the year 2016. Here we have separated 2016 from 2014-
2015 because of different results noted during this time.
The images reveal that the most consistent and strongest source in the city is the SSLF.
This is most evident from the 40+ ppb $CH_4$ enhancements detected at towers 7, 10 and 11 coming
from the location of the SSLF (by triangulation). Enhancements from the landfill appear to also
be detectable at towers 2, 4, 5, and 13. Based on these observations it can be concluded that there
is no other source in Marion County comparable in strength to the SSLF. A small fraction of the
SSLF plume is likely due to the co-located wastewater facility, but the inventory estimates suggest
that the wastewater treatment facility is responsible for no more than 7% of this plume (Cambaliza
et al., 2015; Massakkers et al., 2016). The PEP, located in the northwestern section of the city,
may be partially responsible for a plume of 5-10 ppb at towers 5 and 11. However, the plume is
less detectable using the criterion 2 background value that has higher background (using tower 8



as a background) from NW wind direction (not shown), adding uncertainty to the true magnitude
of the enhancement from this source. The same is true for towers 2 and 13, which have pronounced
plumes when winds are from the NW with the criterion 1 background, but when background 2 is
used these plumes vanish (not shown). Such inconsistency makes it difficult to attribute these
plumes to an urban source.

Another important point is the cluster of large enhancements surrounding tower 10 in 2014

- 2015. Because no other tower sees these enhancements (at least at comparable magnitudes), we
believe that these plumes are the result of local NG leaks likely from residential sector of
Indianapolis. These plumes are not consistent temporally or spatially as they mostly disappear in
2016, potentially indicating that they are transient NG distribution leaks. It is reasonable to
hypothesize that NG related $CH_4$ is being emitted by diffuse, small leaks all across the city.
However, towers downwind of the city do not see a large or distinct enhancement from the city,
especially when compared to the SSLF source. Thus, the diffuse NG source suspected to be twice
as large as the SSLF source (Lamb et al., 2016) does not appear to be supported by these data.
This finding contradicts conclusions made by Cambaliza et al., (2015), who attributed most of the
$CH_4$ emitted by Indianapolis to NG related activities. We hypothesize that the relatively high
Indianapolis $CH_4$ emissions (see Fig. 1) reported by Cambaliza et al., (2015) are the result of the
low sample size of airborne flux estimates, which is prone to large random errors (see section 3.2).
Our results indicate that the main $CH_4$ source in the city is SSLF and that other sources potentially
associated with NG distribution are difficult to identify with clarity. This conclusion is in
agreement with EPA 2012 inventory (section 2.3).

**4 Conclusions**



We have examined four specific contributions to discrepancies between urban top-down and
bottom-up $CH_4$ emission estimates from Indianapolis; domain definition, heterogeneous
background mole fractions, temporal variability in emissions, and source knowledge. Results
indicate that the urban domain definition is crucial for the comparison of the emission estimates
among various methods. Atmospheric inverse flux estimates for Marion County, which is similar
to the domain that is analyzed by inventory and airborne mass balance methodologies (Mays et
al., 2009, Cambaliza et al., 2014, Lamb et al., 2016), is 107 mol/s compared to 160 mol/s that is
estimated for the larger domain (Hestia inventory domain). This partially explains higher
emissions in inverse modeling estimates shown by Lamb et al., (2016); however, 107 mol/s is still
about 40-50% higher than what EPA and Lamb et al., (2016) find in their inventories (Fig. 1).
The midday Indianapolis atmospheric $CH_4$ mole fraction background is shown to be
heterogeneous with 2-5 ppb, statistically significant biases for NW, W, SW, S and SE wind
directions. We focus on midday atmospheric conditions to avoid the complexities of vertical
stratification in the stable boundary layer. Background random error is a function of sample size
and decreases as a number of independent samples increase. Low sample volumes, such as a few
hours of data from a single location, are prone to random errors on the order of 10-20 ppb in the
$CH_4$ enhancement, similar to the magnitude of the total enhancement from the city of Indianapolis.
Longer-term sampling and/or more extensive background sampling is necessary to reduce the
random errors. Several days of measurements (e.g. 25 total hours of measurement) would reduce
random errors to 3-5 ppb, noticeably smaller than the typical enhancement from Indianapolis
emissions. This large random error in the $CH_4$ background may explain Heimburger et al. (2017)
finding of large variability in airborne estimates of Indianapolis $CH_4$ emissions. Given many



samples, the airborne studies converge to an average value of $CH_4$ flux that is close to inventory
estimates for Indianapolis (see Fig. 1).

Measurement and analysis strategies can minimize the impacts of these sources of error.

Spatially extensive measurement of upwind $CH_4$ mole fractions are recommended.  For towers or
other point-based measurements, multiple upwind measurement locations are clearly beneficial.
For the aircraft mass balance approach, we recommend an upwind transect be measured, lagged
in time if possible, to provide a more complete understanding of the urban background conditions.
Complex background conditions might suggest that data from certain days or wind directions
should not be used for flux calculation.  Finally, a mesoscale atmospheric modeling system
informed with the locations of important upwind $CH_4$ sources can serve as a powerful complement
to the atmospheric data (Barkley et al., 2017).  Such simulations can guide sampling strategies,
and aid in interpretation of data collected with moderately complex background conditions.

With regard to temporal variability, no statistically detectable changes in the emission rate

were observed when comparing 2014 and 2016 $CH_4$ emissions.  However, a large difference
between day and night $CH_4$ emissions was implied from a simple budget estimate.  Night (20-5
LST) emissions may be 2 times lower than the emissions during the afternoon (12-16 LST) hours.
Because prior estimates of top-down citywide emissions are derived using afternoon-only
measurements, overall emissions of Indianapolis may be lower than these studies suggest.  This
bias may be present in studies performed in other cities as well.  Our study suggests that day/night
differences in $CH_4$ emissions must be understood if regional emission estimates are to be
calculated correctly.  Long-term, tower-based observations are an effective tool for understanding
and quantifying multi-year variability in urban emissions.



One final point addressed in this study is the location of major CH$_4$ sources in Indianapolis.
Analysis of the INFLUX observation data suggests that inventories for Indianapolis are mostly
accurate and that there is likely no evidence of a large, diffuse NG source of CH$_4$ as implied by
Lamb et al., (2016).  The only major source in the city is SSLF and it is observed at multiple
towers.  There is evidence for occasional NG leaks, but they appear localized and limited in their
strength.
Overall, assessment of the CH$_4$ emissions at Indianapolis highlights a number of
uncertainties that need to be considered in any serious evaluation of urban CH$_4$ emissions.  These
uncertainties amplify for Indianapolis since its CH$_4$ emissions are comparable in magnitude to the
regional background flow and as our results show it may be difficult at times to distinguish noise
in the background from the actual city emissions signal.  The evaluation of larger CH$_4$ sources may
be easier with respect to separating signal from background.  However, all of the points raised in
this work will be nonetheless relevant and need to be addressed for our understanding of urban
CH$_4$ emissions to significantly improve.

**Author Contribution**
Nikolay Balashov, Kenneth Davis, and Natasha Miles developed the study and worked together
on generating the main hypothesis of this work.  They also wrote most of the manuscript.  Nikolay
Balashov wrote all of the codes and performed the analyses presented in this work as well as
generated all of the figures.  Natasha Miles and Scott Richardson helped with maintenance and
gathering of the INFLUX tower data.  They also wrote section 2.2 of the paper.  Thomas Lauvaux
helped with the analysis presented in Figure 1 and section 3.1 concerning interpretation of the
inversion modeling results from Lamb et al. (2016).  Zachary Barkley significantly contributed to





discussions regarding the hypothesis and careful presentation of sections 2.6 and 3.3. Timothy
Bonin provided all of the lidar data and wrote the second part of section 2.3 regarding the lidar and
the methodology used to determine planetary boundary layer heights. He also contributed to
sections 2.6 and 3.3.

**Competing Interests**
The authors declare that they have no conflict of interest.

**Acknowledgements**
This research has been supported by the National Institute of Standards and Technology (project
number 70NANB10H245). We would like to thank Dr. Bram Maasakkers for the helpful
discussion regarding the EPA 2012 inventory and the relevant error structure. We also thank Dr.
Paul Shepson and Dr. Dennis Lamb for their useful input regarding airborne mass balance flights
and the process of compiling an emissions inventory.

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





## Tables

**Table 1.** INFLUX towers used to estimate $CH_4$ background based on two different criteria. Numbers in bold indicate towers chosen to generate a background field when multiple options are possible (for more details see discussion). In short, Criterion 1 uses towers with the lowest mean $CH_4$ for a specific wind direction, and Criterion 2 uses towers not downwind of large sources (including the city as a whole).

| Wind Direction | $CH_4$ Background Towers | |
| --- | --- | --- |
| | Criterion 1 | Criterion 2 |
| North (N) | 8 | **13**, 8 |
| Northeast (NE) | 8 | **13**, 8, 2 |
| East (E) | **2**, 8 | **8**, 4, 1, 2 |
| Southeast (SE) | 1 | **8**, 13, 4, 1 |
| South (S) | 1 | **4**, 13, 1 |
| Southwest (SW) | 13 | **1**, 4 |
| West (W) | 1 | **4**, 1 |
| Northwest (NW) | 1 | **8**, 1 |



**Figures**

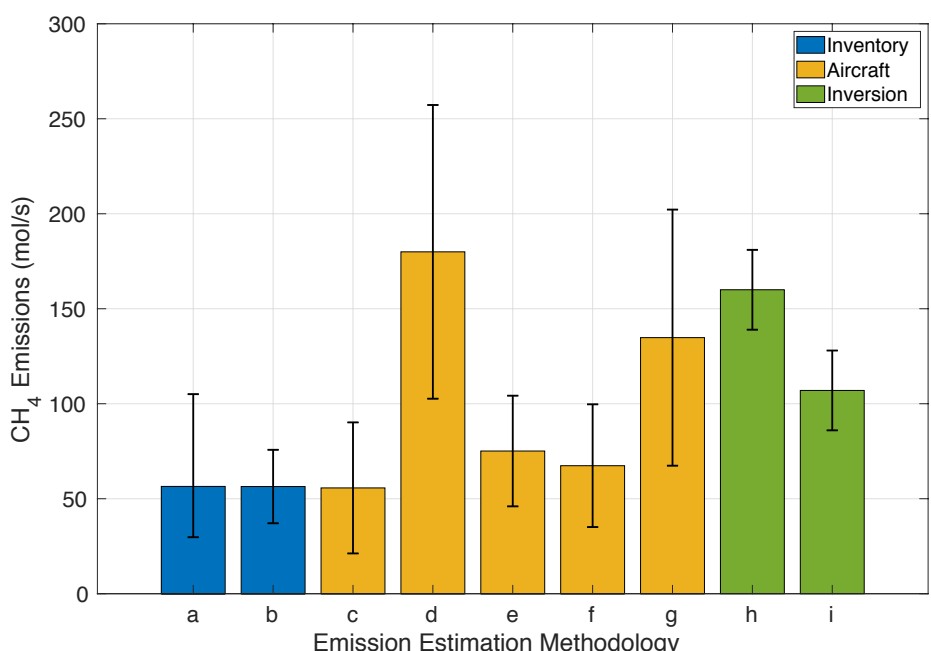


**Figure 1.** Various estimates of CH$_4$ emissions at Indianapolis. **(a, b)** Bottom-up estimates of CH$_4$ emissions
conducted by Lamb et al. (2016) in 2013 and Maasakkers et al. (2016) based on the EPA 2012 inventory
respectively. Error bars show 95% confidence intervals (for more details see above-mentioned articles).
**(c-g)** Top-down evaluations of CH$_4$ emissions with aircraft from various flight campaigns where **(c)**
contains 5 flights over March-April of 2008, **(d)** contains 3 flights over November-January of 2008-09, **(e)**
contains 5 flights over April-July of 2011, **(f)** contains 9 flights from November-December, 2014, and **(g)**
contains the same 5 flights over April-July of 2011 but uses different methodology. Methodologies for **(c-f)** are described in Lamb et al. (2016) and methodology for **(g)** is described in Cambaliza et al. (2015).
Error bars show 95% confidence intervals (for more details see above-mentioned articles). **(h, i)** Top-down
evaluations of CH$_4$ emissions for 2012-2013 using tower inversion modeling methodology with two
different domains, where **(h)** uses the full domain of Fig. 3 and **(i)** uses only the Marion County domain of
Fig. 3. The inversion methodology and 95% confidence intervals are described in detail in Lamb et al.
(2016).





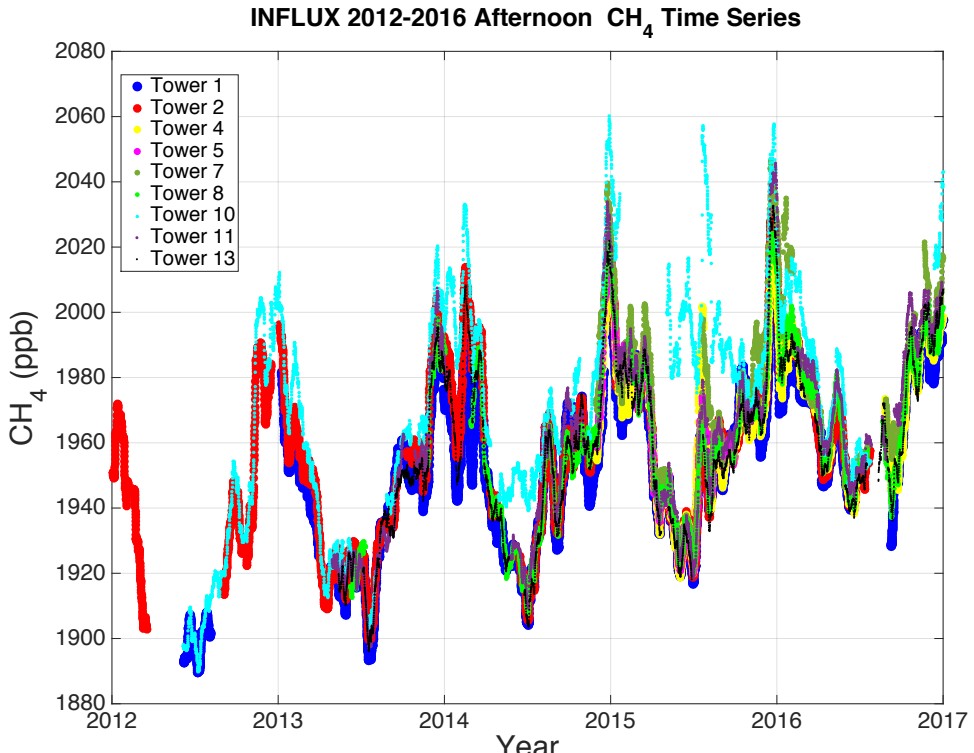


**Figure 2.** 20-day running average of afternoon (12-16 LST; the hours are inclusive) CH$_4$ mole fractions
as measured by the INFLUX tower network (highest available height is used) from 2012 through 2016.



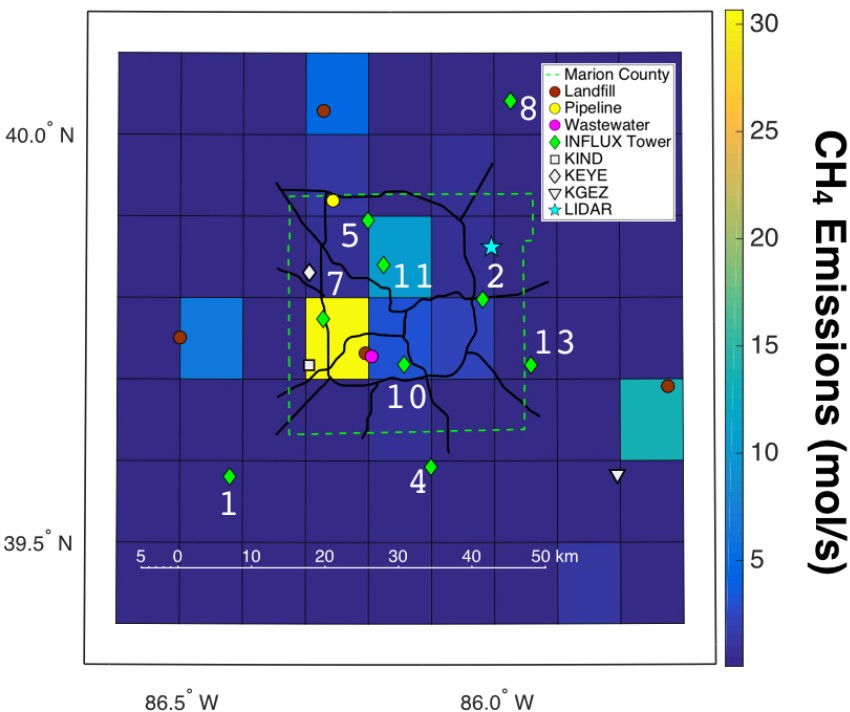

**Figure 3.** Map of the primary roads in Indianapolis, INFLUX towers, lidar system, weather stations, and a few $CH_4$ point sources plotted over the gridded $CH_4$ emissions (mol/s) from the EPA 2012 Inventory (Maasakkers et al., 2016). The gridded map of emissions includes emissions from these point sources; their position is provided to aid in interpretation of the observations. The dashed bright green line denotes Marion County borders.



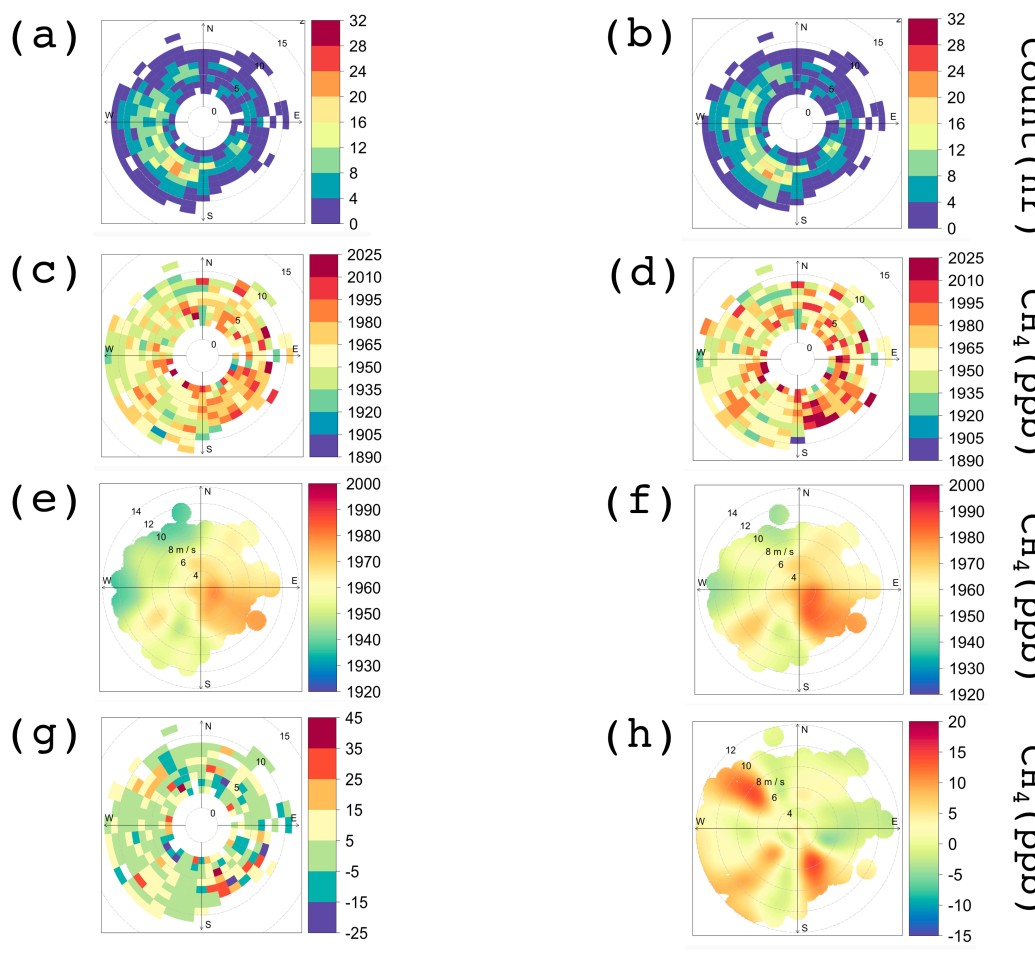

**Figure 4.** Frequency and bivariate polar plots of $CH_4$ background for Indianapolis using data from 12-16 LST, and November, 2014 through December, 2016 given 2 different criteria (Table 1). **(a)** Polar histogram of number of daily measurements available using criterion 1. **(b)** Same as (a) only for criterion 2. Differences between (a) and (b) are due to slight differences in data availability at the considered towers. **(c)** Polar frequency plot of the mean $CH_4$ background using criterion 1. **(d)** Same as (c) only for criterion 2. **(e)** Polar bivariate plot of $CH_4$ background using criterion 1. **(f)** Same as (e) only for criterion 2. **(g)** Polar frequency plot of difference between the backgrounds: $criterion\ 2 - criterion\ 1$. **(h)** Same as (g) but shown with a bivariate polar plot.



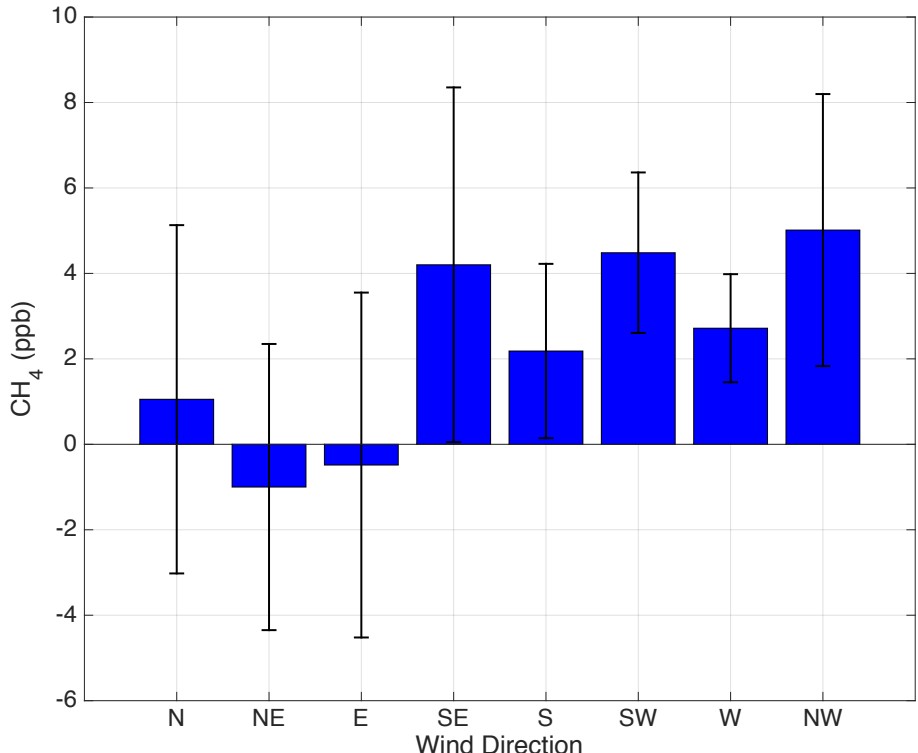

**Figure 5.** Average of the differences between criteria 2 and 1 $CH_4$ backgrounds at Indianapolis as a function
of wind direction. These averages are generated from the same data as used in Fig. 4 and reflect results
shown in Fig. 4g. Error bars indicate standard error × 2.



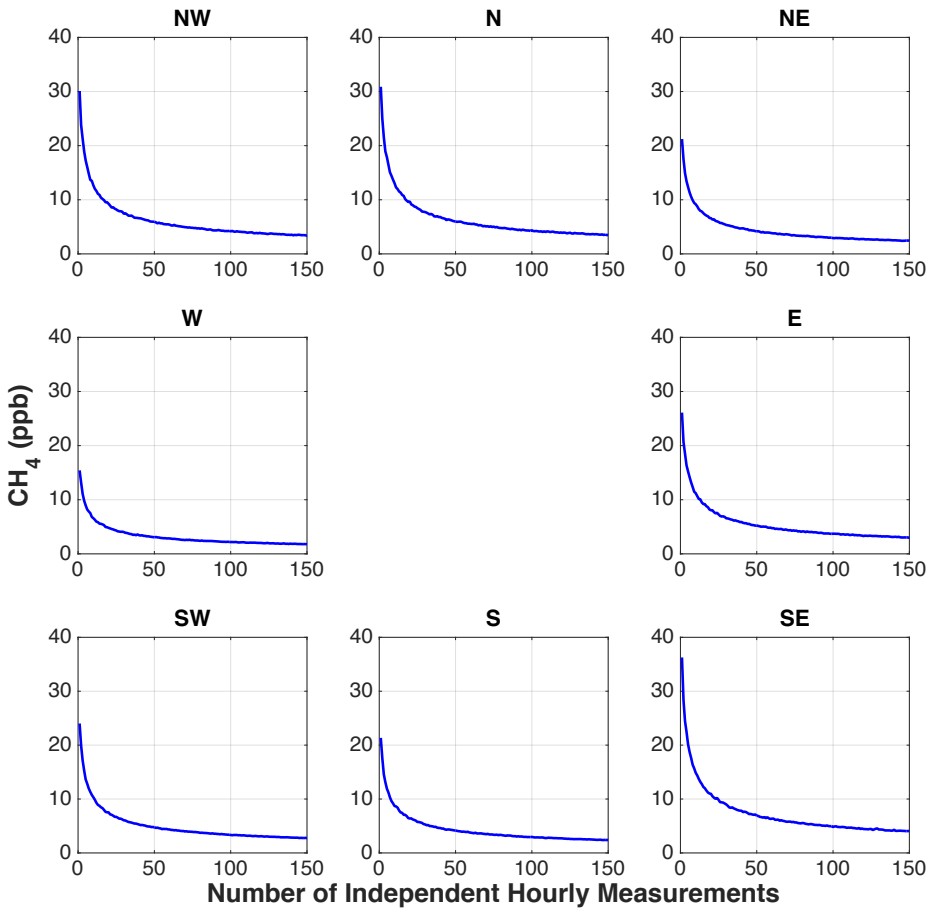

**Figure 6.** Bootstrap simulation of the standard errors multiplied by 2 in Indianapolis $CH_4$ background mole
fraction differences (between criteria 2 and 1) as a function of sample size and wind direction (see text for
details).





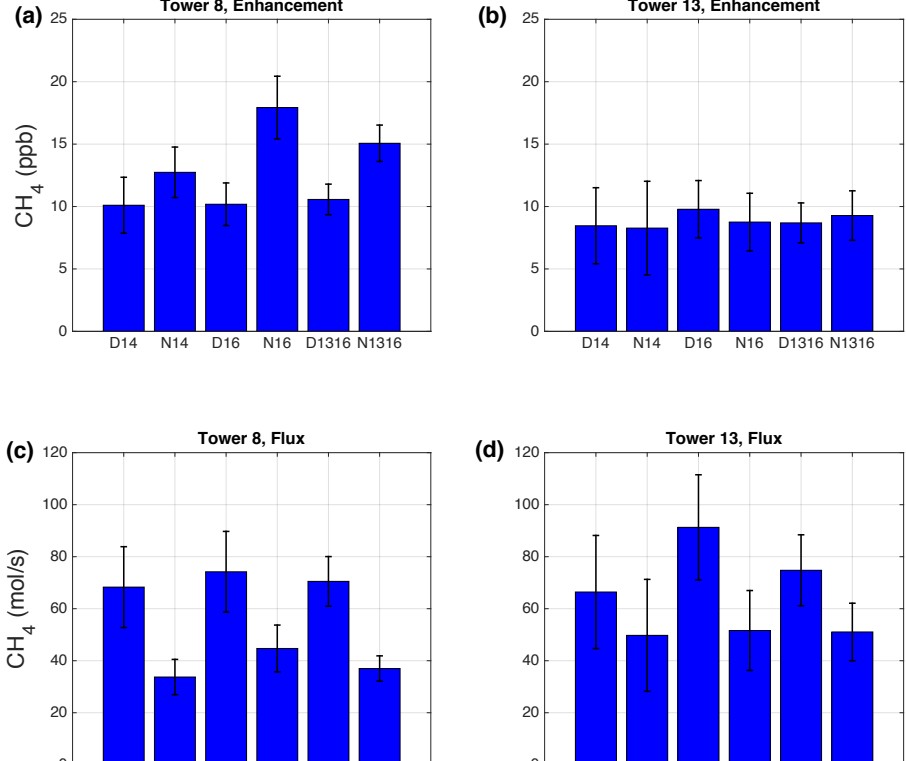

**Figure 7.** Averages of the daytime (D) and nighttime (N) $CH_4$ enhancements and fluxes at INFLUX towers 8 and 13 for years 2014 (14), 2016 (16), and 2013-2016 (1316). The error bars represent 95% confidence interval of each mean value. **(a)** Estimates of $CH_4$ enhancements from tower 8. **(b)** Estimates of $CH_4$ enhancements from tower 13. **(c)** Estimates of $CH_4$ flux from tower 8. **(d)** Estimates of $CH_4$ flux from tower 13.



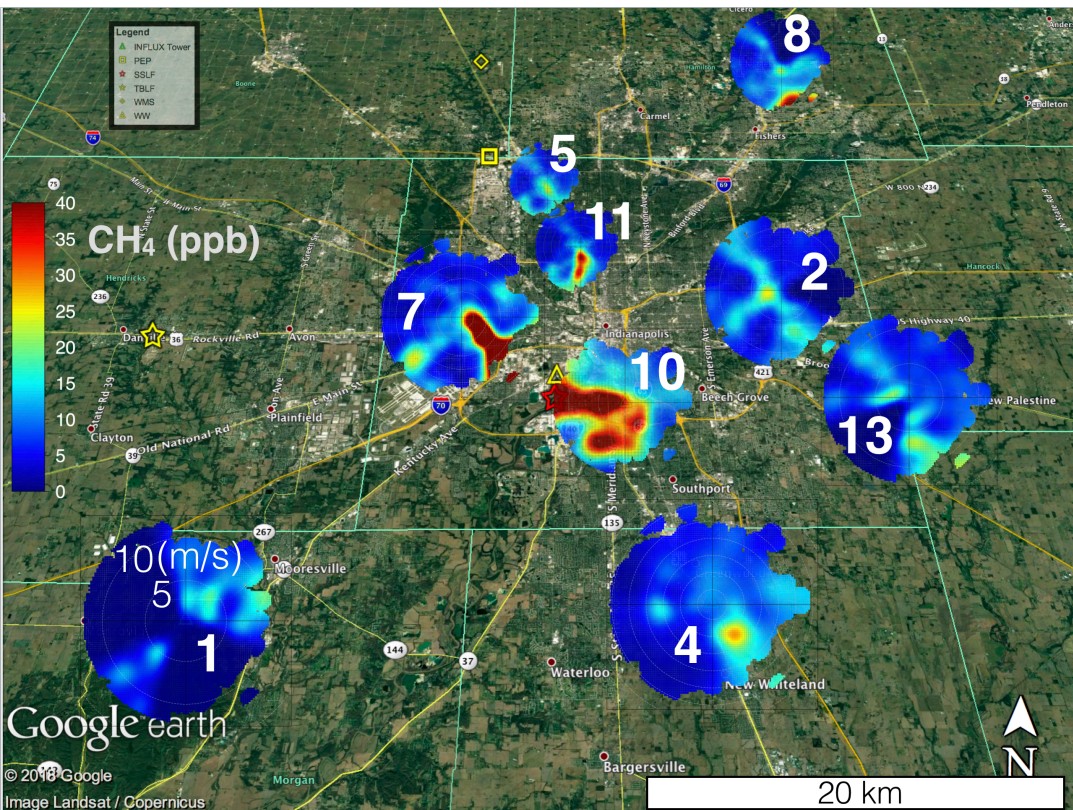

**Figure 8.** Google Earth image overlaid with bivariate polar plots (section 2.5) of the $CH_4$ enhancements at 9 INFLUX towers in Indianapolis using the criterion 1 background (Table 1) for full years of 2014 and 2015 over the afternoon (12-16 LST). The wind speed scale is only labeled at site 1; other sites follow the same convention. Legend indicates known sources of $CH_4$: Panhandle Eastern Pipeline (PEP), Southern Side Landfill (SSLF), Twin Bridges Landfill (TBLF), Waste Management Solutions (WMS), Waste Water treatment facility (WW). The known magnitudes of sources that are in Marion County (PEP, SSLF, and WW) are reported in section 2.3. Magnitudes of TBLF and WMS according to EPA are approximately 5 mol/s. The largest known source on the map is SSLF.




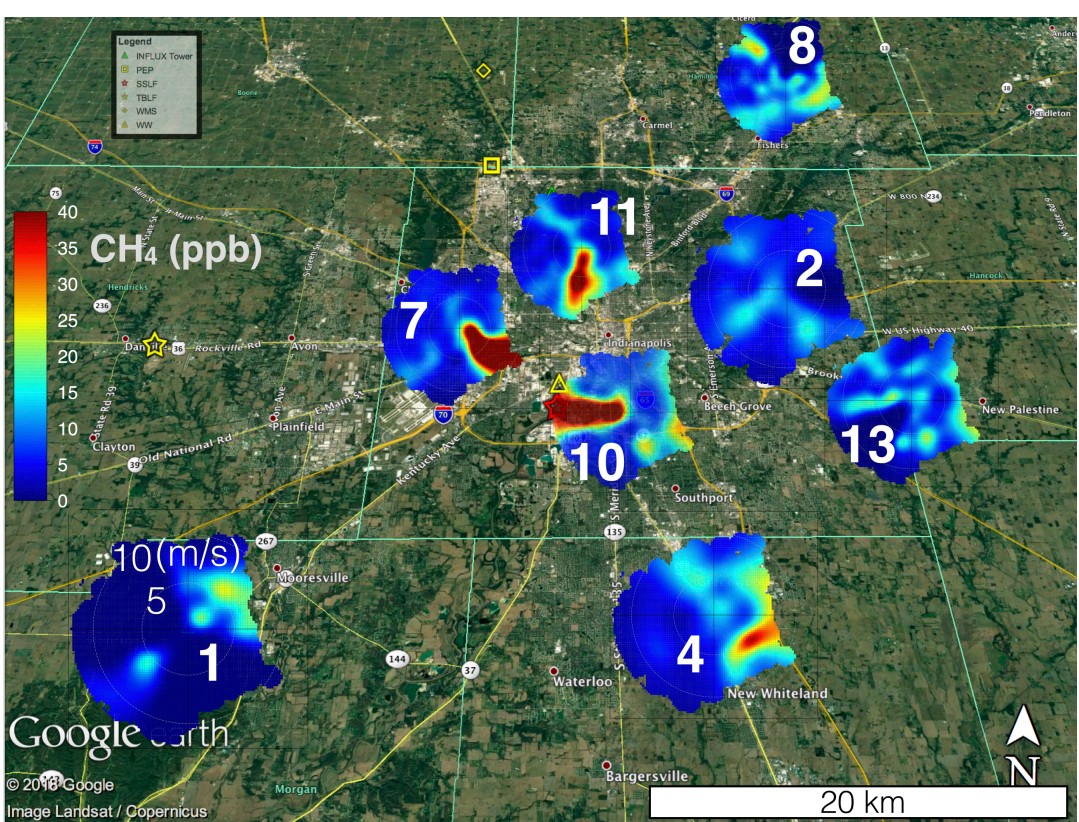

805

**Figure 9.** Same as Fig. 8 only for the year of 2016.