# Peer review of "Background Heterogeneity and Other Uncertainties in Estimating Urban Methane Flux: Results from the Indianapolis Flux (INFLUX) Experiment"

_Atmospheric Chemistry and Physics, 2019_

## Referee Comment (RC1) · Anonymous Referee #2 · 5 Apr 2019

Balashov et al., investigate components of an inverse modelling framework and how these may affect estimates of methane emissions. This is a well written paper with clearly described methods and results. The conclusions drawn are reasonable and are highly applicable to those working on city-scale and mesoscale inversions of methane and other tracers.

Methane concentrations are on the rise, and there is still a great deal of uncertainty about where these emissions are coming from. It is important to investigate how sensitive estimates of methane emissions are to the method and assumptions, and investigate which approaches are most reasonable. The INFLUX experiment resulted in a high density of in situ measurements, both tower and flight measurements, which provides a unique platform to test approaches for determining city-level emissions of methane through methods such as inverse modelling. Specifying the spatial domain, the approach to dealing with background concentrations, the periodicity of the measurements and temporal variability in these measurements, and unknown sources of methane missing from the bottom-up approaches are all important considerations in determining city-level emissions.

There are a few sections which may require some clarification, and it may be helpful to the reader if the authors reordered some of the content.

The order of the subsections in section 2 (Methods) does not follow the order that the four components of the inverse modelling framework are listed in the abstract or discussion section. It may make it easier for the reader to follow from the methods through to the results if there is an explicit methods section for each of the four sensitivity analyses.

Section 2.4 lines 172 to 185: Could the authors explain what the percentages are referring to and provide the overall quantity? Does this come from an inventory analysis?

The section following, starting at line 187, is also labelled as Section 2.4.

Section 2.5 line 218: Should this be a subsection of the previous background section (2.4)? Or should this subsection be called Variability in Background Concentration?

Equation 3 line 250: This equation and notation are taken from Hanna et al. 1982. It may be useful to the reader to explain the terms more thoroughly. Most of the explanation for terms is taken from Chapter 9 of this text, but some of the terms are explained more fully in earlier chapters. I was particularly distracted by the $z_i$ until I realised that it was conventional for the $z_i$ to denote the boundary layer height of the box, rather than $i$ as an indicator for height at time step $i$ (or distance $i$). Do you take $z_i$ to be the average

boundary layer height during the hour?

In the text the units of Qa are described to be in mass per unit time per unit area, and the units of C are described to be in mole fraction in the first instance. Should the concentration not be converted to moles per volume? Later on from line 263 this does appear to be the case.

Section 3.1 lines 292 to 307: This information seems to be related to methods, and should perhaps be in Section 2, probably before the methods section on Background concentrations to be consistent with the abstract and discussion sections.

Line 350: I think it would be useful to justify why you needed to use the bootstrapping approach.

Lines 350 to 363 and Figure 6: I would propose that the analysis for assessing the number of measurements required to obtain a reliable background concentration estimate should focus on the standard deviation rather than on the standard error.

It is unsurprising that the standard error (standard deviation of the mean) reduces as the sample size is increased, as you state in line 359. This is a property of the standard error. If you're interested in the stability of the background concentration difference estimate, you should rather be looking at the standard deviation of the concentration differences (which you can do in the same way, except instead of looking at the standard deviation of the mean values for the 5000 samples, you look at the mean of the variances for the 5000 samples – which can then be converted to a standard deviation). The standard deviation will provide information about the uncertainty in the background concentration. The plots in Figure 6 are slightly different for each wind direction because the standard deviation of the background concentration from these directions differs and because the bias for each direction differs. At some optimum sample size the standard deviation of the concentration differences will stabilize for each direction. If you assume that under normal circumstances (where you don't have two competing domains or multiple background sites) you would be able to obtain the uncertainty in

the background concentration from historical data and this uncertainty remains stable over time, you would want to ensure that the background concentration estimate is obtained from a sufficiently large sample size so that you know what the uncertainty in this estimate is. Therefore, if you have multiple background sites where you can assess at which sample size the standard deviation of differences between background sites stabilizes, you may want to determine for each wind direction what this sample size is and what the stabilized standard deviation is. Basing the background concentration on a sample of this size or larger should provide an estimate with a predictable uncertainty, which is now independent of the sample size. If you know what the standard deviation is, then it follows what sample size is required to obtain a background concentration with the required standard error (precision) (if you can use SE = SD/sqrt(n)).

If you decide to stick with the standard error plots, I think you should show on each plot at which sample size each direction reaches the required precision, say 3 ppb, as this differs for each wind direction.

A point that should be discussed is that the measurements you obtain for the background site are taken at different times, and as the number of measurements increases, so too does the averaging period, which changes the interpretation of this average. There's a danger that if the averaging period is too long, the background concentration measurements may be representative of different synoptic periods.

Line 453-454: "Background random error is a function of sample size and decreases as a number of independent samples increase". As mentioned earlier, I don't think that this is the interesting part of the temporal variability analysis and is already a property of standard error. I think it would be more interesting to discuss how different the standard deviation in the background concentration differences are between wind directions. It would be interesting to know if the sample size at which the standard deviation of the background concentrations stabilizes is similar for all directions. This would be a helpful number if it can be assumed to be generalizable. Basing the minimum sample size on the standard error is less generalizable as the required precision may differ, and the

[Figure]

variability in background concentrations would differ between regions.

Figure 9: This caption should be expanded in order to make each figure stand-alone.

---

## Referee Comment (RC2) · Anonymous Referee #3 · 22 Jul 2019

This paper looks at methane emissions from the city of Indianapolis and explores some of the reasons that the many previous studies on the topic have yielded very different emission estimates using different methodologies. This paper is a useful and relevant contribution to the topic of atmospheric inverse methods for urban methane emissions quantification, but I have a couple of complaints.

The narrative of the paper would be improved by more directly addressing the relationship between this paper and the previous investigations of methane from Indianapolis (all of the papers cited in Figure 1). The Introduction does a fine job of giving a general

review of the previous studies, but the connection to the central problem of different studies/methods yielding different results is weaker in the rest of the sections. For example, how are the methods for background estimation in this paper different than the methods used in the previous studies? In the Conclusions, you recommend a strategy for background estimation for the aircraft mass balance method, but you did not describe how it was done in the previous studies. Also, the Lamb paper identifies a major discrepancy between top-down and bottom-up estimation of the non-biological portion of Indianapolis methane emissions, and the current paper is a follow-on to that paper, but it is not clear whether this paper resolves that question or not, or only partly resolves it.

Specific Comments:

Line 19 – details about the type of analyzers and the measurement heights in the abstract are unnecessary and irrelevant.

Line 103: Please describe briefly how/why the landfill emissions are considered well-known.

Section 2.2 – There are extra details here that are not very relevant to the current paper and have already been described in other papers. This section could be made more concise.

Section 2.4 (Sources) – Although references are given, the source of the numbers in this paragraph is not clear. Are they from direct measurements, an inventory, or something else?

Section 2.4 (Background) – As written, I had to read this section many times to try and understand it and I'm still not sure I fully understand the two methods, so it needs to be re-worked for clarity. Why is a viable method not to take the lowest measurement among all towers at a given hour as a background? How do these two approaches compare to those used in the cited aircraft and tower-based top-down studies?

Section 2.5 – How far away are your receptors and wind measurement locations since you say that this method requires them to be nearby?

Section 2.6, Line 261 – Which towers were used for which wind directions?

Section 3.1 - The first three sentences could probably be condensed into one concise sentence without losing any meaning. - I am having trouble squaring your description of the domain differences with my understanding of the Lamb paper. By my reading, the Lamb paper describes developing an inventory for the larger domain, but you say that the inventory covered mostly only Marion County. I find it hard to believe that Lamb et al. would perform such a detailed analysis and accidentally compare totally different areas for the inventory and tower inversion. What am I misunderstanding? - With the revised inverse emissions estimate, it is not clear whether you've replicated the methods of the inversion in Lamb et al. over a smaller domain, or whether you've used the boundary layer budget method described in the method section.

Section 3.2 - How much of the data are filtered using the criteria you give? - Line 372: Suggest: "Because Indianapolis is a relatively small emitter of methane, and because there are relatively large sources outside of the city, uncertainties due to background estimation are comparatively large."

Section 3.4 - Isn't the result that the SSLF is the largest and strongest source in the city consistent with your prior understanding, as described in the methods? - Looking at figure 8, I can't tell which enhancements around T10 you think are from SSLF and which are from NG leaks. - How do you square your findings that emissions from NG is not a significant portion of emissions with the findings in Lamb et al. that approximately half the emissions are from NG using ethane as a tracer? Are you saying that you can't see the sources because they are below your detection threshold or that their signals or swamped by that of the SSLF, or are you saying that their existence is entirely not supported by the data? On the one hand, you say there is not much evidence for a diffuse NG source, but on the other hand, even after adjusting the domain, your top-down

estimate is still much higher than the inventories. You should at least acknowledge this contradiction or remaining possible existence of unknown sources. - Line 488: The description of "occasional" on seems incorrect since this apparent signal shows up in Figure 8, which represents a two-year average.

Technical Comments: Line 61: Suggest: ". . . atmospheric methods and inventory assessment have sometimes succeeded. . ." Are there are cases when these two criteria have been met but reconciliation has not been achieved? Line 70: Suggest: "Recent studies of urban CH4 emissions in California indicate. . ." Line 72: The phrasing "large NG infrastructures" is strange and evokes large individual pieces of equipment, which I don't think is your intent. Line 79: Suggest: "in" → "for" Line 85: Suggest: ". . .comprised of irregular or periodic in situ aircraft measurements, continuous in situ observations. . ." Line 91: Suggest: "well-suited" → "designed" Line 94: Delete: "Recently" Line 103: Suggest: "Uncertainty in total emissions is driven by. . ." Line 132: Suggest: "tubes secured" → "air collected" Line 139: Suggest: "inflow" → "sample air" Line 152: The given link re-directs to some other website. Line 154: Suggest: "The anemometers are located at about 10 m AGL." Line 200: Suggest: ". . .based on two different sets of criteria. Both approaches identify. . ." Line 316: Suggest: "inventory" → "inventories (Fig. 1)" Line 352: The meaning of the numbers "2 to 150" is unclear. Line 385: Suggest: "at least twice as high" → "approximately twice as large" Line 396: Suggest: "did not change significantly between 2014 and 2016." Line 519: "Dennis" – Do you mean Brian?
* * *

---

## Author Comment (AC1) · 9 Sep 2019

**Response to Referee # 2**

We would like to thank the reviewer for their helpful remarks. Below we address all of the comments presented to us by the reviewer.

*There are a few sections which may require some clarification, and it may be helpful to the reader if the authors reordered some of the content.*

*The order of the subsections in section 2 (Methods) does not follow the order that the four components of the inverse modelling framework are listed in the abstract or discussion section. It may make it easier for the reader to follow from the methods through to the results if there is an explicit methods section for each of the four sensitivity analyses.*

The sections are now slightly reordered to better reflect the whole structure of the paper.

*Section 2.4 lines 172 to 185: Could the authors explain what the percentages are referring to and provide the overall quantity? Does this come from an inventory analysis?*

These are percentages of the total Indianapolis methane emissions depending on a given estimation. For instance, some estimates (Cambaliza et al., 2015) found landfill to contribute only to 22% of the total Indianapolis methane emissions, while other estimates (GHG reporting program) found landfill to be responsible for 63% of total city methane emissions. The total emission value could be the same or different. Some estimates come from inventories; others come from top-down studies such as aircraft mass balance and inversion. Figure 1 in the article summarizes different estimates of methane at Indianapolis. It just does not have the breakdown of the total emissions by sources. We know the breakdown only for some studies. There is now an attempt to summarize this breakdown in section 2.7.

*The section following, starting at line 187, is also labelled as Section 2.4.*

This is an error. It is now fixed.

*Section 2.5 line 218: Should this be a subsection of the previous background section (2.4)? Or should this subsection be called Variability in Background Concentration?*

Section 2.5 describes bivariate plots, so it is titled correctly. Due to the reordering of the methodology part of the paper some sections now changed their number.

*Equation 3 line 250: This equation and notation are taken from Hanna et al. 1982. It may be useful to the reader to explain the terms more thoroughly. Most of the explanation for terms is taken from Chapter 9 of this text, but some of the terms are explained more fully in earlier chapters. I was particularly distracted by the $z_i$ until I realised that it was conventional for the $z_i$ to denote the boundary layer height of the box, rather than $i$ as an indicator for height at time step $i$ (or distance $i$). Do you take $z_i$ to be the average boundary layer height during the hour?*

Yes, $z_i$ is confusing variable name, so we have changed it to H. Yes, we assume H represents boundary layer height for a given hour. We are thinking to leave the rest of the description as is (some minor edits are incorporated), but if there is anything that you would like us to explain specifically we could do that.

*In the text the units of $Q_a$ are described to be in mass per unit time per unit area, and the units of C are described to be in mole fraction in the first instance. Should the concentration not be converted to moles per volume? Later on from line 263 this does appear to be the case.*

You are absolutely correct. It was an error; it should say concentration in the first instance. On line 263 it states that because our data is given in mole fractions we must first convert it to concentration (mass per volume) before we can apply the budget equation. Correct, we are not converting $CH_4$ to moles per volume, but to complete abovementioned conversion from $CH_4$ mole fractions to $CH_4$ concentration we do need to use average molar dry density of air, which has units of moles per volume.

*Section 3.1 lines 292 to 307: This information seems to be related to methods, and should perhaps be in Section 2, probably before the methods section on Background concentrations to be consistent with the abstract and discussion sections.*

Lines 299 to 307 have been moved to methods as requested. The other lines seem to fit appropriately to the results section as they give the domain issue some context. It would be to jarring to jump into line 308 right away.

*Line 350: I think it would be useful to justify why you needed to use the bootstrapping approach. Lines 350 to 363 and Figure 6: I would propose that the analysis for assessing the number of measurements required to obtain a reliable background concentration estimate should focus on the standard deviation rather than on the standard error.*

Bootstrapping approach allows us to vary sample size of a theoretical experiment and to see how the background uncertainty responds to it. Below I explain why this may not work for the standard deviation.

*It is unsurprising that the standard error (standard deviation of the mean) reduces as the sample size is increased, as you state in line 359. This is a property of the standard error. If you're interested in the stability of the background concentration difference estimate, you should rather be looking at the standard deviation of the concentration differences (which you can do in the same way, except instead of looking at the standard deviation of the mean values for the 5000 samples, you look at the mean of the variances for the 5000 samples – which can then be converted to a standard deviation). The standard deviation will provide information about the uncertainty in the background concentration. The plots in Figure 6 are slightly different for each wind direction because the standard deviation of the background concentration from these directions differs and because the bias for each direction differs. At some optimum sample size the standard deviation of the concentration differences will stabilize for each direction. If you assume that under normal circumstances (where you don't have two competing domains or multiple background sites) you would be able to obtain the uncertainty in the background concentration from historical data and this uncertainty remains stable over time, you would want to ensure that the background concentration estimate is obtained from a sufficiently large sample size so that you know what the uncertainty in this estimate is. Therefore, if you have multiple background sites where you can assess at which sample size the standard deviation of differences between background sites stabilizes, you may want to determine for each wind direction what this sample size is and what the stabilized standard deviation is. Basing the background concentration on a sample of this size or larger should provide an estimate with a predictable uncertainty, which is now independent of the sample size.*

*If you know what the standard deviation is, then it follows what sample size is required to obtain a background concentration with the required standard error (precision) (if you can use SE = SD/sqrt(n)).*

*If you decide to stick with the standard error plots, I think you should show on each plot at which sample size each direction reaches the required precision, say 3 ppb, as this differs for each wind direction.*

Thank you for such a thoughtful comment. This really gets me thinking deeper about this background variability concept.

I agree with you that it is unsurprising that standard error is decreasing with increasing sample size. However, I do find Figure 6 useful in that standard errors vary depending on wind direction. So given a sample size of around 20 we can get a general idea which wind direction will provide us with the lowest random error of the difference between backgrounds. It is important to note that this Figure 6 does not say anything at all about bias. This is just random error. The bias is a difference between backgrounds and is shown in Figure 5. In ideal world we would want this difference to be zero and have very little random error on it. Unfortunately that is not the case here. The background is quite complex. Now, you do make a valid point regarding standard deviation and that it also gives us information regarding background variability. Because of this I added to Figure 5 another plot that shows standard deviation times 2 for each of the wind directions. That shows us potential background discrepancy that can occur on a given a single day. This is also useful. Here is the updated Figure 5. On the other hand, standard error shows us that as sample size increases our average difference of backgrounds would approach a known bias. But yes, on any given day things could be really variable or not so variable. Additionally, standard deviation plot indicates that W is the best direction regarding the background. It has the lowest variability of background differences. It does have a bias, but overall error is the smallest. This also is evident in Figure 6, where W standard error is the smallest.

[Figure]

[Figure]

**Figure 5.** Average of the differences between criteria 2 and 1 $CH_4$ backgrounds at Indianapolis as a function of wind direction. These averages are generated from the same data as used in Fig. 4 and reflect results shown in Fig. 4g. Error bars indicate in (a) standard error × 2 and in (b) standard deviation × 2.

To respond to your other request, I tried to perform the bootstrap experiment with standard deviations, but it does not seem to work. The variance stabilizes very fast and does not seem to be a function of sample size (maybe only initially). So I think that it would suffice to add a standard deviation plot to Figure 5 because these standard deviations are basically the same ones you would get with the bootstrap experiment. I think this happens because we are sampling from the pool of the same differences and ultimately there is no way for variances to change much after 5000

iterations. In other words, low sample size with 5000 iterations will be similar to a large sample size with 5000 iterations because both of these cases sample from the same PDF of background differences. Please see Figure 6b.

[Figure]

**Figure 6b**. Bootstrap simulation of the standard deviations multiplied by 2 in Indianapolis CH$_4$ background mole fraction differences (between criteria 2 and 1) as a function of sample size and wind direction (see text for details).

To answer your last part of this question, I do not think that we have a required precision at this point for background differences. The plot illustrates at approximate what sample size we would approach a reasonable standard error. Perhaps one way to classify a required precision would be to make sure that the standard error (random error) reaches a point where it is less than Indianapolis enhancement of about 12 ppb (a higher estimate of the Indianapolis enhancement from section 3.2) by a factor of 2 when combined with a bias (in this case it is helpful to think in terms of absolute magnitudes, so let say the requirement is 6 ppb). In this case each wind direction would

have a different threshold. For instance, given this requirement NW direction would need a random error of 1 since its bias is 5. For NW direction, this threshold would require more than 150 samples. For N direction on the other hand, where the bias is 1, the requirement is fulfilled when random error crosses 5 ppb at 74 samples. This is going to be added to this to the paper now.

Here are the absolute values of the biases for each wind direction, their respective required thresholds, and needed sample size:

N: 1 ppb      T: 5 ppb      N: 74
NE: 1 ppb    T: 5 ppb      N: 36
E: 0.5 ppb    T: 5.5 ppb    N: 46
SE: 4 ppb     T: 2 ppb      N: >150
S: 2 ppb       T: 4 ppb      N: 53
SW: 4.5 ppb T: 1.5 ppb    N: >150
W: 3 ppb      T: 3 ppb      N: 52
NW: 5 ppb   T: 1 ppb      N: >150

[Figure]

**Figure 6a.** Bootstrap simulation of the standard errors multiplied by 2 in Indianapolis CH$_4$ background mole fraction differences (between criteria 2 and 1) as a function of sample size and wind direction (see text for details). Thresholds for each of the wind directions indicate a random error threshold needed for the background uncertainty to be within 50% assuming average CH4 enhancement from Indianapolis is 12 ppb.

*A point that should be discussed is that the measurements you obtain for the background site are taken at different times, and as the number of measurements increases, so too does the averaging period, which changes the interpretation of this average. There's a danger that if the averaging period is too long, the background concentration measurements may be representative of different synoptic periods.*

There are definitely cases when a front is rolling through the area and the methane gradients are huge causing a background difference of 40 ppb or so.

Averaging over a long period of time smooths out these variations. This is why large sample size of estimations is suggested because unavoidably in any type of top-down experiment (aircraft or inversion) there are going to be days that are unsolvable because of complex background. The hope is that there are more days with homogenous background (background variability is less than city enhancement) than days with heterogeneous background (background variability is more than city enhancement). We can improve our chances by eliminating wind directions that are especially problematic.

*Line 453-454: "Background random error is a function of sample size and decreases as a number of independent samples increase". As mentioned earlier, I don't think that this is the interesting part of the temporal variability analysis and is already a property of standard error. I think it would be more interesting to discuss how different the standard deviation in the background concentration differences are between wind directions. It would be interesting to know if the sample size at which the standard deviation of the background concentrations stabilizes is similar for all directions. This would be a helpful number if it can be assumed to be generalizable. Basing the minimum sample size on the standard error is less generalizable as the required precision may differ, and the variability in background concentrations would differ between regions.*

Part of this question is already addressed above. It would be preferable to carry out an analysis presented in this paper for any other region in question where CH4 emissions are of interest. As you point out, each region is unique and presents its own challenges. But if one is able to understand what issues may arise when beginning their top-down estimation for a particular area, they may be able to avoid large errors simply by better constraining their experiment.

*Figure 9: This caption should be expanded in order to make each figure stand-alone.*

Done.

**Response to Referee # 3**

*The narrative of the paper would be improved by more directly addressing the relationship between this paper and the previous investigations of methane from Indianapolis (all of the papers cited in Figure 1). The Introduction does a fine job of giving a general review of the previous studies, but the connection to the central problem of different studies/methods yielding different results is weaker in the rest of the sections.*

We would like to thank the reviewer for their insightful suggestions. We tried to address noted issues to the best of our ability. Each comment of the reviewer is addressed below individually.

*For example, how are the methods for background estimation in this paper different than the methods used in the previous studies?*

This is a very good question. The description of methods to determine background used by other studies is now added to Section 2.4 (methodology section has been reordered due to requests from another reviewer).

The point of the article in question is to emphasize the challenge that background estimation may pose for flux estimation. There is no perfect method for background estimation; it is something that must always be addressed differently depending on a situation at hand. For instance, for aircraft mass balance studies there are 3 main methods to determine a background value. First is to pick a smallest edge value of a transect. Second is to linearly interpolate background field of a transect from one edge to another edge. Third is to use an upwind transect as a background field. In the case of an inversion, it is common to pick a tower that is located generally away from the sources and has on average smallest overall enhancement. In the current paper, background is chosen using 2 different criteria. For both of these criteria, each wind direction (using 8 main wind directions) is considered separately. This is possible because we have 9 towers and we can always change our background tower based on a wind direction. This is an advantage as with this strategy we may be able to better reduce contamination of local sources and to better represent upwind sources that are approaching the city.

*In the Conclusions, you recommend a strategy for background estimation for the aircraft mass balance method, but you did not describe how it was done in the previous studies.*

Here is what we wrote: "For the aircraft mass balance approach, we recommend an upwind transect be measured, lagged in time if possible, to provide a more complete understanding of the urban background conditions. Complex background conditions might suggest that data from certain days or wind directions should not be used for flux calculation." The strategy recommended is intended to better help an analyst to understand the background conditions for a given day. It may be possible that the day in question should not be used for flux estimation due to complex and variable background. We have not seen this type of critical discussion in the articles that employed aircraft mass balance methodology. Generally speaking, there are some "standard" methods of background calculation for an aircraft approach. Unfortunately no method is perfect and each one has its disadvantages. First is to pick a smallest edge value of a transect. This method could be wrong when the upwind plume is narrow and is not represented well by the edges of a transect. Second method is to linearly interpolate background field of a transect from one edge to another edge. This method is better, but it may not always account for the complex gradient of background that may occur over the plume. However, this method could help to identify that background is complex and the day should not be analyzed further for a flux value. Third is to use an upwind transect as a background field. This is potentially the best method if a case is in steady state, but realistically the issue here is lag. Plume is always moving, so upwind and downwind transects are not sampled simultaneously. If typical aircraft mass balance approach assumptions are satisfied this should work well, but from our experience that is not always the case and therefore a closer analysis must determine if a given day is acceptable for a flux estimation or not depending on how background is behaving.

So our goal was not to introduce a new background methodology for aircraft studies necessarily, rather to suggest caution when such data is analyzed. However, we did add the background estimation methodology of aircraft studies as you suggested (Section 2.4) as well as some recommendation regarding background for aircraft data analysis (conclusion).

*Also, the Lamb paper identifies a major discrepancy between top-down and bottom-up estimation of the non-biological portion of Indianapolis methane*

*emissions, and the current paper is a follow-on to that paper, but it is not clear whether this paper resolves that question or not, or only partly resolves it.*

In this paper we are unable to address this question directly as we have no measurements of ethane (C2H6), which is a tracer gas used by Lamb et al., 2016 to separate biological CH4 from non-biological CH4. However, we can answer this question indirectly by estimating total emissions of the city and subtracting "known" biological sources (such as landfill, see the comment about landfill for more details) from that total. The residual is hypothesized to originate from non-biological sources such as NG. So we think it is likely there is no major discrepancy between top-down and bottom-up solutions. We think that some of the top-down solutions in Lamb et al. 2016 are biased high and should be lower more in line with bottom-up estimations. As we point out later uncertainty remains, but the high top-down estimates could be potentially explained by the erroneous assumptions in analyzes.

**Specific Comments:**

*Line 19 – details about the type of analyzers and the measurement heights in the abstract are unnecessary and irrelevant.*

The details are now removed.

*Line 103: Please describe briefly how/why the landfill emissions are considered wellknown.*

Originally this statement was based on the Greenhouse Reporting Data (GHGRP), which gave very similar emissions values for this landfill over 2010-2015 time frame. This is supported by Lamb et al. 2016 paper that also cited GHGRP as a reputable source for the landfill. However, 2016 and 2017 GHGRP indicates a 70% decrease in landfill emissions. That seems unrealistic given that our towers near the landfill do not show any decrease in average methane mixing ratios over these years in comparison with the previous years (Figure R1). We are currently in the process of investigating this discrepancy. So far we received no response from EPA regarding this.

[Figure]

**Figure R1.** Yearly CH4 enhancement directional profiles for 4 INFLUX towers located in the city of Indianapolis: (a) tower 7, (b) tower 11, (c) tower 10, and (d) tower 2. Note that there is a lot more variability in the towers closer to the landfill (Towers 10 and 11). Often the plume is unable to fully mix in a close proximity to these towers resulting in the higher variability. Landfill peak is apparent in all of the towers. Tower two, which is more representative of a fully mixed plume, shows no dramatic change in the landfill emissions from year 2015 to year 2016.

For more details see (this needs to be copied and pasted into a browser): https://ghgdata.epa.gov/ghgp/service/facilityDetail/2017?id=1002683&ds=E&et=&popup=true

We do have some potential evidence for variability in landfill emissions from Cambaliza et al. 2015 article that used aircraft mass balance on five different occasions to calculate methane flux from this landfill. Their average is higher than GHGRP, but their estimation has high uncertainty. They were not able to make consistent calculations. Their city totals seem abnormally high on two of the flights, which may indicate there were issues with a background assumption. In addition, Cambliza et al. 2015 used California Landfill Methane Inventory Model v 5.4 or CALMIM to estimate methane emissions from South Side Landfill  (SSLF), where results are more consistent with GHGRP.

Overall, we think we know an approximate value of the SSLF methane emissions, but this section in the paper about the landfill is now rewritten to reflect a truer state of knowledge regarding SSLF landfill emissions.

*Section 2.2 – There are extra details here that are not very relevant to the current paper and have already been described in other papers. This section could be made more concise.*

The section's length has been slightly reduced.

*Section 2.4 (Sources) – Although references are given, the source of the numbers in this paragraph is not clear. Are they from direct measurements, an inventory, or something else?*

The sources of theses numbers are a combination of bottom up and top down estimates to give a range of possible emission values. This section is now rewritten as another reviewer also asked about this.

Here it is:

**2.7 CH$_4$ Sources**

Only a few known CH$_4$ point sources exist within Indianapolis (Cambaliza et al., 2015, Lamb et al., 2016). The Southside Landfill (SSLF), located near the center of the city, is the largest point source in the city with emissions ranging between 28 mol/s (inventory) and 45 mol/s (aircraft) depending on an emission estimation methodology. SSLF could account for as little as 22% or as much as 63% of total Marion County CH$_4$ emissions (top-down from Cambaliza et al., 2015; invetnory from Maasakkers et al., 2016; inventory and top down from Lamb et al., 2016) contigent on how much of the total city emissions are coming from NG. Other city point sources are comparatively small; the wastewater treatment facility located near SSLF contributes about 3-7 mol/s (inventory from Lamb et al. 2016), and the transmission-distribution transfer station at Panhandle Eastern Pipeline (also known as a city gate and further in this study abbreviated as PEP)

is estimated to be about 1 mol/s (inventory from Lamb et al. 2016). The remaining $CH_4$ sources, mainly from NG and livestock, are considered to be diffuse sources and are not well known. Potential sources of emissions related to NG activities include gas regulation meters, emissions from transmission and storage, and Compressed Natural Gas (CNG) fleets. These diffuse NG sources account for 21-69% (this value varies due to the uncertainty in SSLF emissions) of the city emissions or 20-64 mol/s (top down from Cambaliza et al., 2015; inventory from Maasakkers et al., 2016; inventory and top-down from Lamb et al., 2016). Livestock emissions for Marion County are estimated to be around 1.5 mol/s (inventory from Maasakkers et al., 2016). An important question remains of whether SSLF or NG is the dominant CH4 source in Indianapolis. There could also be a possibility of temporal variability in either of the sources as described in the section above.

*Section 2.4 (Background) – As written, I had to read this section many times to try and understand it and I'm still not sure I fully understand the two methods, so it needs to be re-worked for clarity. Why is a viable method not to take the lowest measurement among all towers at a given hour as a background? How do these two approaches compare to those used in the cited aircraft and tower-based top-down studies?*

We apologize for the confusion. Also this section should be numbered 2.5, we will correct the numbering in the next version of the paper.

The main point of this section is that we can pick multiple backgrounds for a given situation and it would be impossible to say which one is better. Hence, two arbitrary but acceptable backgrounds are chosen here to estimate methane enhancements. If background is uniform or closely so, as sometimes stated in literature, then we would see no significant difference between the enhancements calculated with different backgrounds. Yet we show here that this difference is significant and choice of background matters.

We edited this section to try to clarify this point.

It is not a viable method to take the lowest measurement among all towers at a given hour as a background because the background we are interested in is not always the lowest value as illustrated in the schematic shown in Figure R2. True background lies upwind of the city (or the downwind tower of interest from which the enhancement is calculated) and it is not the lowest value. Because we are trying to identify the enhancement specifically from the city we must subtract exactly what is coming into the city. The methane is heterogeneous as described earlier and therefore it is a challenge to identify exact background even at a not-so-large scale as Indianapolis.

[Figure]

**Figure R2.** Theoretical representation of boundary layer $CH_4$ plume heterogeneity across Indianapolis and nearby areas when winds are from the northwestern direction. The colors indicate relative $CH_4$ concentrations where yellow is neutral, blue is low, and red is high. Green dashed lines indicate the assumed boundaries of Indianapolis. Also shown are INFLUX towers with $CH_4$ measurements and known sources.

The approaches used in Criteria 1 and 2 are not entirely different from aircraft and inversion studies, but it is exactly the point. All of these approaches are acceptable, it is just important to understand what kind of uncertainty they are causing. In some cases, we recommend not to perform

flux calculation since the uncertainty is too large due to complex background. Identifying those complex days is whole another topic, which deserves a separate paper, but we do think this is very important and must be emphasized.

Here are a couple of figures that show the heterogeneity of methane in Indianapolis. Indianapolis CH4 observations indicate highly variable background with day-to-day variations at times reaching 150 ppb (Fig. R3). Similarly, WRF-CHEM simulations show occasional spatial non-uniformity of CH4 (Fig. R4).

[Figure]

**Figure R3.** Daily CH$_4$ medians over 15-22 UTC at 9 INFLUX towers.

[Figure]

**Figure R4.** 27 km WRF-CHEM simulations of CH$_4$ enhancements (ppb) for 2 different days using EPA 2012 CH$_4$ emissions (Maasakkers et al, 2016): **(top left)** late morning on 02/07/15 **(top right)** late afternoon on 02/07/15 **(bottom left)** late morning on 02/18/15 **(bottom right)** late afternoon on 02/18/15.

*Section 2.5 – How far away are your receptors and wind measurement locations since you say that this method requires them to be nearby?*

In this case towers are the receptors and they are labeled as green diamonds in the figure below (Fig. R5). The wind data is gathered from the 3 stations identifiable by white colored shapes on the figure as explained in section 2.3. We found that the wind measurements are generally consistent between these 3 stations and therefore their combination is well representative of the city overall. Perhaps occasionally tower 8 may not be represented perfectly by these winds, but we do not think that such situation occurs often.

[Figure]

**Figure R5.** Map of the primary roads in Indianapolis, INFLUX towers, lidar system, weather stations, and a few $CH_4$ point sources plotted over the gridded $CH_4$ emissions (mol/s) from the EPA 2012 Inventory (Maasakkers et al., 2016). The gridded map of emissions includes emissions from these point sources; their position is provided to aid in interpretation of the observations. The dashed bright green line denotes Marion County borders.

*Section 2.6, Line 261 – Which towers were used for which wind directions?*

This question is answered at the end of Section 2.6 (lines 283-287). Here are the relevant sentences: "For S and SW wind directions tower 8 observations are used to represent downwind conditions with background observations coming from towers 1 and 13, respectively (based on Criterion 1 shown in Table 1). For W wind direction, tower 13 observations represent the downwind with background obtained from tower 1. The wind direction is required to be sustained for at least 2 hours, otherwise the data point is eliminated."

*Section 3.1 - The first three sentences could probably be condensed into one concise sentence without losing any meaning. - I am having trouble squaring*

*your description of the domain differences with my understanding of the Lamb paper. By my reading, the Lamb paper describes developing an inventory for the larger domain, but you say that the inventory covered mostly only Marion County. I find it hard to believe that Lamb et al. would perform such a detailed analysis and accidentally compare totally different areas for the inventory and tower inversion. What am I misunderstanding? - With the revised inverse emissions estimate, it is not clear whether you've replicated the methods of the inversion in Lamb et al. over a smaller domain, or whether you've used the boundary layer budget method described in the method section.*

Unfortunately there is indeed an inconsistency between domains used by Lamb inventory and inversion. The domain used by the inversion contains 3 landfills that are not part of the Lamb inventory. We are not sure how this happened, but that is what we are seeing when we read Lamb's paper. The Figure R3 shows domain used by the inversion. You can find this domain in the supplemental of Lamb paper. We also have access to the prior used in the inversion of Lamb paper and these 3 landfills are in there. You can see landfills marked by the brown dot in the Figure R5.

The point we are trying to make here is that it is imperative to be very careful when such comparisons are performed. It may seem obvious that boundaries of emission areas need to be the same when they are compared, but it seems that occasionally such detail can get neglected when analysis considers many other complex parameters. In this we work we would like to emphasize the importance of this initial step.

The revised inversion estimate just shows what would happen if the original inversion had solved for Marion County. We asked the person who did the original INFLUX inversion to rerun his code for the whole region and then just for the Marion County. The result is shown in Figure 1 of the submitted article. Boundary layer budget method is not used for this result. It is used later in the paper to try to understand temporal variability of $CH_4$ flux in Indianapolis. Clarification is now added to section 2.1.

*Section 3.2 - How much of the data are filtered using the criteria you give? - Line 372: Suggest: "Because Indianapolis is a relatively small emitter of methane, and because there are relatively large sources outside of the city, uncertainties due to background estimation are comparatively large."*

The answer is at the beginning of the Section 3.2. Here are the relevant sentences: "To make the comparison as uniform as possible, only data from 12-16 LST are utilized (all hours are inclusive), when the boundary layer is typically well-mixed (Bakwin et al., 1998). A lag 1 autocorrelation is found between 12-16 LST hours, i.e., the hourly afternoon data are correlated to the next hour, but the correlation is not significant for samples separated by two hours or more. Therefore, hours 13 and 15 LST are eliminated to satisfy the independence assumption for hourly samples. Furthermore, we make an assumption that the data satisfy steady state conditions. If the difference between consecutive hourly wind directions exceeds 30 degrees or the difference between hours 16 and 12 LST exceeds 40 degrees, the day is eliminated. Days with average wind speeds below 2 m/s are also eliminated due to slow transport (the transit time from tower 1 to tower 8 is about 7 hours at a wind speed of 2 m/s)."

Because the city of Indianapolis is surrounded be sources that are similar to its CH4 flux magnitude it is not surprising that occasionally there are complex background scenarios that are difficult to address (include modeled map?). If Indianapolis was much larger source than its surrounding sources background would not be a big issue, but in this case it is not so. The goal of our background variability study is to show how variable CH4 background is on average at this location. The data was filtered only to eliminate extreme cases, but other cases, which are likely to be used by inversion studies, or even aircraft methodologies were left in. Another important point of our study is that inversions and mass balances should be carefully filtered to exclude complex background days. However, we did not see an evidence of that filtering in case of studies performed at Indianapolis.

Thank you. Suggestion is incorporated.

*Section 3.4 - Isn't the result that the SSLF is the largest and strongest source in the city consistent with your prior understanding, as described in the methods? - Looking at figure 8, I can't tell which enhancements around T10 you think are from SSLF and which are from NG leaks. - How do you square your findings that emissions from NG is not a significant portion of emissions with the findings in Lamb et al. that approximately half the emissions are from NG using ethane as a tracer? Are you saying that you can't see the sources because they are below your detection threshold or that their signals or swamped by that of the SSLF, or are you saying that their existence is entirely not supported by the data? On the one hand, you*

*say there is not much evidence for a diffuse NG source, but on the other hand, even after adjusting the domain, your top-down estimate is still much higher than the inventories. You should at least acknowledge this contradiction or remaining possible existence of unknown sources. - Line 488: The description of "occasional" on seems incorrect since this apparent signal shows up in Figure 8, which represents a two-year average.*

It is indeed consistent with our prior understanding; however, there are views that natural gas could be larger than SSLF as a source of CH4. We agree that Figure 8 does not provide definitive answer, yet it allows us to see that the landfill is likely the strongest source in the city as no other point or area source is making such a noticeable enhancement at any of the towers. Landfill is located to the west of tower 10 and therefore the plume that arrives from the west is unquestionably belongs to landfill. Even in Lamb et al. 2016 we see similar result by observing low ethane to methane ratios from southwest of tower 11. In general, the ethane to methane ratios provide an insight into the composition of a given plume measured at a certain point, but overall city composition is hard to measure unless the tower is downwind of the whole city. The issue here is that plumes may come outside of the city and we have no way of separating plumes inside the city from the outside using tower 11. Lamb article does say that they ran an inversion of sorts, but unfortunately they poorly explain how they did it and the assumptions that were made in those calculations make the result largely uncertain. We spent sometime trying to understand what they mean by "source footprints" in their supplemental material (S3.4); however, this terminology is not standard and therefore we have no idea how they arrived at their answer. Also their sample size is just 11 days, which is not enough to make a definitive conclusion. And their domain includes sources outside of Marion County (they even mention multiple landfills), which makes this even more confusing.

With regard to adjusted inversion, there is still uncertainty in that result since it only used 3 towers at most (the tower record is sparse in 2012-2013 time frame). In our second attempt of running this inversion it seemed sensitive to the prior, which hinted that the system might need some more experimenting and testing. However, we admit that we cannot say for sure there is no significant diffuse source at Indianapolis comparable to landfill. But the evidence suggests that it is not as large as previously was suggested. The goal of this paper is to show that some of the very large values attained by the top-down estimates in this case appear to be unconvincing since

adjustments in background and domain do affect the flux values. We are going to leave a space for potential diffuse source as suggested, but we must stress that its existence is highly uncertain given the data at this point.

**Technical Comments**

*Line 61: Suggest: "...atmospheric methods and inventory assessment have sometimes succeeded..." Are there are cases when these two criteria have been met but reconciliation has not been achieved?*

This depends on the definition of reconciliation. However, it may be possible that a study found an agreement between an inventory and a top-down methodology, while another study did not find that for the same region. Then perhaps reconciliation is under question.

*Line 70: Suggest: "Recent studies of urban CH4 emissions in California indicate..."*

Done.

*Line 72: The phrasing "large NG infrastructures" is strange and evokes large individual pieces of equipment, which I don't think is your intent.*

Done.

*Line 79: Suggest: "in" → "for"*

Done

*Line 85: Suggest: ". . .comprised of irregular or periodic in situ aircraft measurements, continuous in situ observations. . ."*

Done.

*Line 91: Suggest: "well-suited" → "designed"*

Done.

*Line 94: Delete: "Recently"*

Done.

*Line 103: Suggest: "Uncertainty in total emissions is driven by. . ."*

Done.

*Line 132: Suggest: "tubes secured" → "air collected"*

That is probably okay as it is.

*Line 139: Suggest: "inflow" → "sample air"*

Done.

*Line 152: The given link re-directs to some other website.*

Fixed.

*Line 154: Suggest: "The anemometers are located at about 10 m AGL."*

Done.

*Line 200: Suggest: ". . .based on two different sets of criteria. Both approaches identify. . ."*
Done.

*Line 316: Suggest: "inventory" → "inventories (Fig. 1)"*

Done.

*Line 352: The meaning of the numbers "2 to 150" is unclear.*

These numbers indicate an experiment sample size. Clarification is added. The idea is to see by how much the uncertainty decreases if the sample size is 150 (arbitrary large sample size) values. In theory each value could be used to solve for flux. But with sample size of 2 the uncertainty is large. This is an attempt to try to figure out how much data is optimally needed to solve for the emissions from the city. But due to various assumptions this is just an approximation. Topic related to this has been covered in great depth in the

response to another reviewer. The revised article will contain some changes in that section.

*Line 385: Suggest: "at least twice as high" → "approximately twice as large"*

Done.

*Line 396: Suggest: "did not change significantly between 2014 and 2016."*

Done

*Line 519: "Dennis" – Do you mean Brian?*

Yes, sorry.

---

## Author Response (AR2)

**Response to Referee # 3**

We would like to thank the reviewer for the helpful suggestions. Below we address all of the comments presented to us by the reviewer.

**Reviewer Comments**

*When people talk about "missing" or "unknown" NG sources in urban environments they are usually thinking of leaks that occur downstream of the customer meter. If many small leaks in homes and businesses throughout the city were each leaking a tiny amount, how would that appear on your Figures 6 and 7? Would you be able to see it? The way you describe "unknown" sources you seem to be looking for point sources rather than a broad diffusive source.*

First we would like to apologize regarding our error in figure numbering. We noticed that Figures 8 and 9 somehow got renumbered to 6 and 7 in the main paper. This is now fixed. We assume that this question refers to Figures 8 and 9.

If we are talking about small leaks (what do we mean by small is also an important matter, but we will ignore this for now) such as homes and businesses throughout the city then their signal depends to a large extent on their proximity to towers as well as on the actual amount of these leaks. On Figure 8 it is possible to see some very large enhancements that are occurring near tower 10 (the color scale is saturated, but if we were to increase the color range we would see enhancements near 70 ppb). No other tower sees these enhancements at comparable magnitudes; therefore, it is possible to conclude that these leaks are located near tower 10 and unable to fully mix before they reach our sensor (that is at tower 10). Here immediately we can note an interesting point – these leaks do not appear to be common, at least according to our towers. We do not see anything similar at other towers, and in 2016 these leaks near tower 10 are gone. Of course it is possible that these leaks are unique to some specific infrastructure and perhaps there are still copious small leaks from other places as noted above. In that case, for these leaks to matter they must mix at some point and form a noticeable plume that would emanate from the city. That plume should be at least partially comparable to the landfill plume to really matter. Because if there are small leaks and they are barely detectable it doesn't matter much for the overall city budget. So to address this question a little deeper, let us perform a thought experiment. We can pose the following question: could SSLF explain the SW plume at towers 2 and 8 (these are the towers where the city plume should be generally well mixed)? The reason this question is important is because if landfill can explain this plume, then there is not much space for a significant gas source out there. Sure, there could be a noticeable gas source overall – something like 20-30% of the landfill, but nothing shocking as sometimes implied. In order to address this question it is useful to think of an approximation of a Gaussian plume equation.

The equation could be simplified to the following form:

$$C(x,0,0) = \frac{Q}{U}\frac{1}{\pi\sigma_y\sigma_z} \tag{1}$$

This equation approximates the change in concentration (or mole fraction) of the plume center (of a given gas) as a function of distance away from a source ($x$), where $Q$ is the source strength in mol/s, $U$ is wind speed in m/s, $\sigma_y$ is lateral spread of plume in m, and $\sigma_z$ is vertical spread of plume in m. Both sigmas depend on atmospheric stability and additionally $\sigma_z$ on a height of a
mixed layer. Equation indicates that plume strength will rapidly decline as it moves away from
the source mixing in its surrounding environment. Here we will use this equation to illustrate
how the mole fraction of SSLF declines with distance and how our towers reflect this decline. Of
course what we are about to show is a rough approximation since we are assuming unstable
conditions, winds of 9 m/s, boundary layer of about 700 m, and SSLF strength of 32 mol/s. It is
imperative to understand that these chosen parameters are not so important and general principle
by which plumes spreads remains the same. Occasionally you get higher mole fractions and
occasionally you get lower mole fractions, but over a long period of time these mole fractions
will average out.
Before we return to our thought experiment with Gaussian plume, let us take note of
some interesting aspects of the observations that are shown in Figure 9. Figure 1 (in this
response) shows binned averaged $CH_4$ enhancements by wind direction at towers 10, 11, 2, and 8
for 2016. In some ways this picture is similar to Figure 9 of the article, but allows for a different
perspective. Right away we notice a relatively large enhancements at towers 10 and 11, where
for tower 10 the enhancement is observed when winds are from west (at about 270 degrees) and
for tower 11 the enhancement is observed when winds are from south (at about 180 degrees).
These enhancements occur at wind directions consistent with the location of SSLF. There is a
little doubt that SSLF is the main culprit here. Now if we look at towers 2 and 8, we notice that
tower 2 also exhibits 10-20 ppb enhancement at SW wind direction, which is again consistent
with SSLF. Finally, at tower 8 wind directions of S-SW show a smaller but still noticeable
enhancement of about 10-15 ppb. And again, the direction is consistent with SSLF. Green circles
in Figure 1 emphasize the plumes that are suspected to originate from SSLF. Could these plumes
really be attributed to SSLF?

[Figure]

Figure 1. Directionally (meaning wind direction) averaged $CH_4$ enhancements binned by 10 degrees for
towers 10, 11, 2, and 8 for 2016. Criterion 1 is used for background as described in the article. Green
circles highlight $CH_4$ enhancement that is associated with SSLF.

Let us come back to our thought experiment with Gaussian plume. Figure 2 shows how a
mole fraction of $CH_4$ would vary as we move away from SSLF. The locations of towers 10, 11,
2, and 8 are indicated by the colorful circles. Once again, the equation 1 is a rough estimation of
how plume spreads and not a high-resolution modeling. But on average, plume will generally
follow these rules of dispersion. We are assuming unstable conditions, wind speed of 9 m/s,
source strength of 32 mol/s, and a boundary layer of 700 meters. So once the plume is mixed to
700 meters, it will no longer expand vertically as the layer is capped. At this point the plume
expands only laterally, but that expansion is rather slow as could be noted in the Figure 2. If we
compare mole fractions that are expected by our simple simulation at each of the towers we
notice that they are generally consistent with what is observed in Figure 1. More importantly, the
ratios by which the mole fractions decrease as we move away from SSLF are consistent. In
Figure 1, the plumes peak near 70, 50, 20, and 10-15 ppb at towers 10, 11, 2, and 8 respectively.
This is similar to what Figure 2 shows. The only discrepancy that is noticeable is at tower 11,
which is about 20 ppb higher in Figure 1 than in Figure 2, but that could be explained by an error
of our approximation. There are many factors to which real plume is slightly sensitive as we
described above. The main story however is not changed. The plumes observed in Figure 1 are
closely consistent with what we would expect to originate from a SSLF.  Therefore, it appears
that SSLF is indeed the most important source in the city and is responsible for most of the $CH_4$
emitted. It could be noted that there are other smaller peaks in Figure 1, which indeed are likely
to originate from natural gas leaks. But from this analysis it is clear that these leaks, even if combined, are smaller than the emissions from Landfill. If these leaks were to emit more gas
than a landfill, we would see much larger enhancements at towers 2 and 8, which would be
inconsistent with the scenario where the main source of $CH_4$ in Indianapolis is SSLF. But we do
not observe that. SSLF is able to explain the majority of $CH_4$ that is observed at our towers. Of
course, this Gaussian plume experiment is a crude estimation and we cannot completely rule out
significant source from natural gas leaks, but at this point it is very difficult to make a logical
argument that even if such source does exits that it is bigger than SSLF.

[Figure]

Figure 2. Simulation of plume dispersion using equation 1. The plume originates from SSLF and the
source strength is assumed to be 32 mol/s, which is consistent with EPA estimates. Color circles indicate
towers based on the distance away from SSLF. Y-axis shows mole fractions that are expected at these
towers given a rough approximation resulting from the equation 1.

So to answer your question, the small leaks should appear relatively large on Figures 8
and 9 of the article when they are in close proximity to towers and smaller when they are
detected by the away towers such as 2 and 8. At towers 2 and 8 we could expect leaks to
combine into a single combined plume and if this plume would compare in magnitude to SSLF
we would unquestionably detect it. In fact, we are currently working on another INFLUX paper
where we analyze various flights that measured $CH_4$ downwind of Indianapolis and simulate
these downwind plumes with 3-km WRF-Chem. Preliminary results show that downwind plume
is indeed sensitive to even a slight increases in small gas leaks. Many flights we analyzed do
appear to indicate that SSLF is the main source consistent with the observations, but we will not
get into these results here. It is pointed out here just to support the point that mixed plumes are
well detected by the receptors downwind of the city (such as tower 8 for instance). And yes,
when we describe "unknown" sources, we are talking about both diffuse sources and point sources. For example, if there was another point source comparable to SSLF in the vicinity of
Indianapolis, our tower analysis would immediately pick it up. So just because there is not a
large unknown source in the area this time doesn't mean it is always true for a given city. And if
such source would suddenly appear, we would see it in our towers because of their superb
temporal resolution. Diffuse source will also be detected as already was pointed out in this note
earlier. We recommend our type of analysis to anyone starting to analyze either $CH_4$ or $CO_2$ for a
particular city. It is helpful to understand the observations well before carrying out any kind of
modeling.
Making a small aside here, it is important to point out that a good inversion study should
always carefully examine available data (such as our paper has done, which is a very thorough
analysis of almost every single measurement made at towers) and see if their prior assumptions
make sense. And after inversion is carried out, it is imperative to assess the results and see if they
make actual sense given the observed data. Unfortunately, many studies do not do that. Many
presentations do not even show the raw data used for inversions (such as time series, etc.), let
alone examine the observations with great scrutiny. Now, we are not saying that Lamb et al.
(2016) was not a careful study, but there are certainly aspects in that work that are questionable.

*Section 2.6 in part describes how data are filtered for this analysis but it does not say how many*
*data are remaining. I'm wondering if enough data are available to do seasonal-level analysis?*
We do not actually perform seasonal-level analysis. We do try to understand annual variability of
$CH_4$ emissions by comparing three different time periods. For tower 8 if we counted every single
hour used for daytime analysis, we get 259 values (60 days) for 2014, 346 values (78 days) for
2016, and 756 values (172 days) for 2013-2016. For nighttime analysis, we get 334 values (42
days) for 2014, 306 values (44 days) for 2016, and 789 values (107 days) for 2013-2016. Note
that more hours are allowed per *24-hour-day* for nighttime than daytime (see section 2.6).
For tower 13 if we counted every single hour used for daytime analysis, we get 120
values (31 days) for 2014, 159 values (38 days) for 2016, and 399 values (98 days) for 2013-
2016. For nighttime analysis, we get 94 values (13 days) for 2014, 228 values (31 days) for
2016, and 445 values (64 days) for 2013-2016.
Overall, the number of samples is robust according to the central limit theorem.
*Section 2.7 – Lamb et al. 2016 gives an estimate for the SSLF based on plume dispersion*
*modeling that would seem worth including since it an independent metric based on direct*
*measurements. That paper says the estimates for SSLF from GHGRP, aircraft mass balance and*
*plume dispersion modeling all yield similar estimates. Whereas in this section you give a wider*
*range for the SSLF, citing essential the same sources as Lamb et al. 2016.*
You are correct. Lamb et al. 2016 does perform an inversion to estimate emissions from SSLF
and the result is similar to GHGRP. We will include that estimate. However, estimates of
GHGRP and aircraft mass balances (the ones performed in Cambaliza et al. (2015)) do not yield
similar results hence in our article we point that out. Lamb et al. (2016) does say that new aircraft
estimates of landfill are consistent with GHGRP, but that actual "new" experiment is nowhere to
be found. It is likely that Cambaliza et al. (2015) background estimates are wrong, but Lamb et
al. (2016) did not show how they modified the methodology and therefore it seems not appropriate to use their value over Cambaliza et al. (2015) value. So in our paper we now mention both values: from Cambaliza et al. (2015) and recalculation from Lamb et al. (2016).

*The paragraph starting at line 361 is confusing. Presumably errors were thoroughly assessed in the original work. Are you saying that you are not sure if doing the inversion for the smaller domain is valid? What kind of sensitivity analysis do you think is needed? Same comment for the sentence in the conclusions section starting at line 518. It makes it sound like you have reason to doubt this result, but you haven't said what the reasons are other than "inversions can have large errors."*

You make a good point that this paragraph is not well supported by the data, so we will delete it. Due to personal communications with some of the authors of Lamb et al. (2016), there is a reason to think the inversion errors were not fully assessed. Unfortunately, it is unclear how it could be relayed here, so these passages will have to be deleted. The point was to emphasize that inversion errors are bigger than the error bars suggest in Lamb et al. (2016).

Inversion for smaller domains could be valid if proper meteorological model is used. For one, it is unclear whether the meteorological model used in Lamb et al. (2016) inversion was able to handle boundary layer heights properly. Another issue was prior, which was based on 5 flights from Cambaliza et al. (2015). Cambaliza et al. (2015) should not be used as an estimate for prior due to its poor background strategies and extremely low sample size.

Multiple steps must be performed to insure that inversion is valid. First, footprints derived from meteorological model must be double-checked. To do so derived boundary layer heights must be compared to observations and the corresponding error needs to be known. The analysis of footprints showed that realistic time series at certain towers could not be reproduced. Second issue is prior. How does the inversion vary as prior varies? This analysis was partially performed with the inversion model used in Lamb et al. (2016) and the results were unsatisfactory.

So here it is, yes, there are reasons to doubt inversion. But because it is not possible to put unofficial results into the paper, we will edit the current article to reflect what is published so far.

**Reviewer Suggestions**

We would like to thank the reviewer for carefully reading our manuscript. The suggestions presented here are well-thought-out and noticeably improve this article.

*Lines 25-26: Suggest: "…within the error range of one of the two inventories."*

Done.

*Line 27: Suggest: "…higher than inventory estimates."*

Done.

*Line 28: Suggest: "…spatially heterogenous and temporally variable. Variability in background mole fractions observed at any given moment and a single location could be…"*

Done.
*Line 31: Suggest: "from most wind directions" to "for most wind directions"*
Done.
*Line 35: Suggest: "No significant unknown CH4 sources are found." to "We found no evidence*
*for significant CH4 point sources that are otherwise missing from the inventories."*
Done.
*Line 37: Suggest: "Other sources, such as leaks from the natural gas (NG) distribution system,*
*are localized and transient, and do not appear to be a consistently large source of CH4*
*emissions in Indianapolis." to "Leaks from the NG distribution system that were detected with*
*the tower network appeared localized on non-permanent and do not appear to constitute as large*
*of a source of CH4 as previously hypothesized by some top-down studies"*
Done.
*Line 39: Suggest deleting the sentence: "However some uncertainty…"*
Done.
*Line 51: Delete ", however, "*
Done.
*Line 59: Miller et al. is not the correct citation for the EPA and EDGAR inventories.*
Corrected.
*Line 72: Suggest: "For example, two studies (McKain, Hendrick) indicate that…"*
Done.
*Line 76: Suggest: "due to fugitive emissions from NG infrastructure in urban environments."*
Done.
*Line 96: Suggest: "estimates" to "quantification"*
Done.
*Line 97: Delete "estimates"*

Done.
*Line 99: "which are" to "specifically"*
Done.
*Line 109: Suggest: "from Cambaliza et al. (2015) which used"*
Done.
*Line 113: Delete: "This uncertainty has not yet been resolved."*
Done.
*Line 118-119: Suggest for clarity: "(1) inconsistent geographic boundaries between top-down*
*and bottom-up studies, (2) heterogeneity in the urban scale CH4 background and (3) temporal*
*variability in urban emissions, both of which were not accounted for in top-down studies, and*
*(4)…"*
Done.
*Line 138: Suggest: "has been" to "was"*
Done.
*Line 160: CH4 scale information is irrelevant.*
Deleted.
*Line 163: Suggest: "was" to "were"*
Done.
*Line 168: Suggest: The accuracy of the wind speed measurements are…"*
Done.
*Line 181: Suggest: "...when the determined BLH is unreliable…"*
Done.
*Line 197: Suggest: "Aircraft mass balance studies of Indianapolis mentioned used two main…"*
Done.
*Line 244: Suggest: "...slow transport across the city"*

Done.
*Line 275: "upwind" to "downwind"?*
Yep, should be downwind.
*Line 326: Suggest deleting "contigent on how much of the total city emissions are coming from*
*NG."*
Deleted.
*Line 333: NG distribution leaks are not mentioned as potential sources.*
It is now added into the sentence.
*Line 338: Suggest: "An important question remains of whether SSLF or NG is the dominant*
*CH4 source in Indianapolis, or whether they are approximately equal."*
Done.
*Line 340: "...as described above" – not clear where you are referring to. Could just delete this*
*last sentence.*
Deleted.
*Line 379: "enveloping" to "flowing into"*
Done.
*Lines 434, 439: Figure 7 to 5*
This is actually correct; the Figure number should be 7 here. We apologize for our error in figure
numberings. We fixed it now.
*Line 458: consistent with the larger error bar of Lamb et al. (2016) inventory calculation.*
Done.
*Line 475: Suggest: "Based on these observations it can be concluded that there are no other*
*point source in Marion County comparable in size to the SSLF.*
Done.
*486: Suggest: "an urban" to "a specific"*
Done.

*Line 501: Suggest deleting: "it is imperative…", so it reads, "However, our flux estimates…"*

Done.

*Line 508: Suggest: "We have examined four potential contributions to discrepancies between urban top-down and bottom-up estimates of CH4 emissions from Indianapolis: domain definition, heterogeneous background mole fractions, temporal variability in emissions, and sources missing from inventories."*

Done.

*Line 523: "towers are selected" to "estimates"*

Done.

*Line 532: "Low sample volumes" to "small sample sizes"*

Done.

*Line 542: "noticeable smaller than" to "which is half the magnitude of"*

Done.

*Line 549: "than their individual components as" to "than several of the individual estimates"*

Done.

*Line 580: "since its CH4 emissions are comparable in magnitude to" to "since the enhancement signal from its CH4 
[revised manuscript text omitted]

Nikolai Balashov 12/7/2019 6:24 PM

---

## Author Response (AR3)

**Response to Referee # 3**

We thank the reviewer for providing provocative and helpful comments.

**Reviewer Comments**

*The paper has improved through the review process, but I still have a significant concern that the authors draw conclusions to eliminate the possibility of a large underestimated NG source that are not fully supported by the evidence in the paper. On the topic of whether or not there is evidence for a significant missing NG source in Indianapolis, the authors seem to be posing the wrong the question. The relevant question is not whether NG or the SSLF is the largest source in Indianapolis. No one disputes that there is a large landfill source in the middle of Indianapolis. The question which is relevant to the existing literature and society is whether NG emissions are significantly underestimated by the inventory. Can you reject the hypothesis that the 40% difference between top-down and bottom-up that remains after the domain issue is fixed is due to NG leaks? This question needs to be clearly and consistently addressed in the text. At a minimum, I would like to see the following statements in the paper revised to accurately and consistently represent the evidence given.*

We do not have a fundamental disagreement with the reviewer on these points.

We had some errors in our percentage calculations, which are now fixed.

We cannot "reject the hypothesis that the 40% (it is 91% actually) difference between top-down and bottom-up that remains after the domain issue is fixed is due to NG leaks." We did not intend to make this point, and we have clarified the text to make this clear.

We do conclude that the discrepancy between top-down and bottom-up is less than that suggested by the Lamb et al., (2016) paper. We also do show evidence for the lack of large, persistent point sources other than the SSLF. We have revised the text to clarify these conclusions, and to make sure that these are not conflated.

We also conclude that, given the large potential for random error in the background (one main focus of this manuscript), we cannot yet be confident in the significance of the difference between the inventory and the atmospheric estimate of the diffuse NG source. We have gone through the text to make this point clear where appropriate.

*Abstract, Line 43: "Leaks from the natural gas distribution system that were detected with the tower network appeared localized and non-permanent and do not appear to constitute as large of a source of CH4 as previously hypothesized by some top-down studies." It sounds like you are providing evidence to reject the findings of previous top-down studies, but you are actually only providing evidence for the absence of point sources rather than a diffuse source comprised of many small point sources. This sentence needs rephrasing to more fairly represent the evidence given in the paper.*

Now we say:
The data from the towers confirm that the strongest $CH_4$ source in Indianapolis is South Side
Landfill.  Leaks from the natural gas distribution system that were detected with the tower
network appeared localized and non-permanent. Our simple atmospheric budget analyses
estimate magnitude of the diffuse NG source to be 70% higher than inventory estimates, but
more comprehensive analyses are needed.
*Line 463: Please give a number for "somewhat"*
Now we say:
"If we assume that SSLF emissions are generally known (GHG reporting program) that would
indicate that emissions from NG distribution are likely to be about 14 mol/s (70%) higher than
what both of the inventories currently estimate but within the error bars of Lamb et al., (2016)'s
inventory calculation."
*Line 500: "Thus, the diffuse NG source suspected to be twice as large as the SSLF source (Lamb*
*et al., 2016) does not appear to be supported by these data." This is true for large point sources*
*but not for a broad diffuse source. Please reword or add a sentence to reflect such a possibility.*
Here is the reworked part of that paragraph:
"None of the individual leaks appears to be similar in magnitude to the emissions that originate
from SSLF.  Diffuse NG emissions comparable to the SSLF source (Lamb et al., 2016) may
exist.  Our flux estimations at towers 8 and 13, however, imply that the magnitude of NG diffuse
source suggested by the top-down analyses in Cambaliza et al. (2015) and Lamb et al. (2016) are
probably overestimates (see section 3.3).  We hypothesize that the relatively high Indianapolis
$CH_4$ emissions (see Fig. 1) reported by Cambaliza et al. (2015) could be a result of random errors
in upwind conditions (see section 3.2) influencing the small number of airborne flux estimates."
*Line 574: "Analysis of the INFLUX observation data suggests that inventories for Indianapolis are*
*mostly accurate and that there is no clear evidence of a large, diffuse NG source of CH4 as*
*implied by Lamb et al. (2016)." I take issue with the "mostly accurate" phrasing since you still*
*have a significant difference between top-down and bottom-up (Line 369).*
Changed to:
"Analysis of the INFLUX tower observations suggest a diffuse NG source that exceeds both of
the inventory estimates by 70%, but additionally our analysis shows that the discrepancy is less
than that proposed by highest values reported in Lamb et al. (2016) (see Fig. 1).  Uncertainty
remains regarding the magnitude of the diffuse NG source of $CH_4$.  The only major point source in the city is SSLF and it is observed at multiple towers.  There is an evidence for occasional
point-source NG leaks, but they appear to be transient in time, and limited in their strength."
*Other comments:*
*Abstract, Line 47: I do not understand the meaning of "real". Please rephrase.*
Now it says:
"Long-term averaging, spatially-extensive upwind mole fraction observations, mesoscale
atmospheric modeling of the regional emissions environment, and careful treatment of the times
of day are recommended for precise and accurate quantification of urban $CH_4$ emissions."
*Line 27: Suggest rewording "…and (4) the presence of unknown CH4 sources." –> "…and (4) CH4*
*sources that are not accounted for in the inventory" or "…and (4) CH4 source types that are*
*absent from the inventory"*
Done.
*Line 30: Suggset rewording "…about 35% and thereby lessens the discrepancy by bringing total*
*city flux within the error range of one of the two inventories." -> "…about 35%, thereby*
*lessening the discrepancy and bringing total city flux within the error range of one of the two*
*inventories."*
Done.
*Line 125: Not clear what "both" refers to*
Changed to:
"…temporal variability in urban emissions, which is not captured by the existing top-down
studies…"
*Line 139-140: suggest: "and boundary layer depth compared to nearby rural areas…"*
Done.
*Line 494: Why do you think the plumes are from the residential sector? State your evidence or*
*else remove.*

[revised manuscript text omitted]

---

## Author Response (AR4)

**Response to Editor**

We thank the editor for the robust examination of our paper and useful comments.

*Comments to the Author:*
*Thank you for addressing the issues raised.*

*I have some minor comments below, all about wording.*

*46 ...estimate the magnitude...*
*53 you might want to add a reference to the recent paper by Hmiel et a*
*(https://doi.org/10.1038/s41586-020-1991-8 ).*
*149 ...that, based on the EPA gridded inventory (Maasakkers et al, 2016), increase...*
*170 Wind speed and direction were measured...*
*273ff Temporal variability in what? Background concentration, emission?? Do not write much*
*but do clarify.*
*337 Inventory*
*422 ...using the INFLUX network, we estimate that... (2 changes)*
*444 ...features in Figure 7 is a ...*
*512 ....methods. The atmospheric inverse flux..... (Could be Our atmospheric....)*

All of the comments above were addressed to the best of our ability (see the updated combined document).

[revised manuscript text omitted]